# Deployment and evaluation of an $NH_4^+/H_3O^+$ reagent-ion switching chemical ionization mass spectrometer for the detection of reduced and oxygenated gas-phase organic compounds

Cort L. Zang[1] and Megan D. Willis[1]

[1]Department of Chemistry, Colorado State University, Fort Collins, CO, USA

**Correspondence:** Megan D. Willis (megan.willis@colostate.edu)

**Abstract.** Reactive organic carbon (ROC) is diverse in its speciation, functionalization, and volatility, with varying implications for ozone production and secondary organic aerosol formation and growth. Chemical ionization mass spectrometry (CIMS) approaches can provide in situ ROC observations and the CIMS reagent-ion controls the detectable ROC species. To expand the range of detectable ROC, we describe a method for switching between the reagent-ions $NH_4^+$ and $H_3O^+$ in a Vocus chemical ionization time-of-flight mass spectrometer (Vocus-CI-ToFMS). We describe optimization of ion-molecule reactor conditions for both reagent-ions, at the same temperature, and compare the ability of $NH_4^+$ and $H_3O^+$ to detect a variety of volatile organic compounds (VOCs), semi-volatile, and intermediate volatility organic compounds (S/IVOCs) including oxygenates and organic sulfur compounds. Sensitivities are comparable to other similar instruments (up to $\sim 5 \ \mathrm{counts \ s^{-1} \ ppt_v^{-1}}$) with detection limits on the order of 1-10 s of $\mathrm{ppt_v}$ (1 s integration time). We report a method for characterizing and filtering periods of hysteresis following each reagent-ion switch and compare use of reagent-ions, persistent ambient ions, and a deuterated internal standard for diagnosing this hystersis. We deploy $NH_4^+/H_3O^+$ reagent-ion switching in a rural pine forest in central Colorado, US, and use our ambient measurements to compare the capabilities of $NH_4^+$ and $H_3O^+$ in the same instrument, without interferences from variation in instrument and inlet designs. We find that $H_3O^+$ optimally detects reduced ROC species with high volatility, while $NH_4^+$ improves detection of functionalized ROC compounds, including organic nitrates and oxygenated S/IVOCs that are readily fragmented by $H_3O^+$.

## 1 Introduction

Tropospheric aerosol formation, oxidant reactivity, and ozone production are driven by the molecularly diverse pool of atmospheric reactive organic carbon (ROC; all organic species excluding methane) (Heald and Kroll, 2020). Speciation of atmospheric ROC is an ongoing analytical challenge (e.g., Goldstein and Galbally, 2007; Hunter et al., 2017), especially at time resolutions relevant to atmospheric mixing and chemistry. While reduced volatile organic compounds (VOCs, with saturation vapor concentration, $C^*$, $> 3 \times 10^6 \ \mathrm{\mu g \ m^{-3}}$) are an important fraction of ROC, functionalized species with lower volatility (semi-volatile and intermediate volatility organic compounds, S/IVOCs with $C^*$ between 0.3 and $3 \times 10^6 \ \mathrm{\mu g \ m^{-3}}$) are major contributors to ozone production and aerosol formation (e.g., Xu et al., 2021; Heald and Kroll, 2020; Bianchi et al., 2019; Donahue et al., 2011). For example, near comprehensive measurements of ROC at a forested site showed that S/IVOCs contribute

approximately one third of ·OH-reactivity and potential secondary organic aerosol production (Hunter et al., 2017). Further, semi-volatile and oxygenated VOCs contribute to marine secondary aerosol formation (Burkart et al., 2017; Mungall et al., 2017; Croft et al., 2019, 2021), and oxygenated species such as furans contribute significantly to ·OH-reactivity and aerosol production in wildfire plumes (Xu et al., 2021). In many urban environments, volatile chemical products and other classes of IVOCs make a growing contribution to aerosol and ozone production (Coggon et al., 2021; Zhao et al., 2014b). However, owing to limitations in analytical techniques, and partitioning to inlet and instrument surfaces (Deming et al., 2019; Pagonis et al., 2019), oxygenated and otherwise functionalized S/IVOCs are often unmeasured.

Chemical ionization mass spectrometry (CIMS) represents a family of analytical techniques applied to detect and characterize organic and inorganic trace gases in whole air at high time resolution (e.g., Zhang et al., 2023; Yuan et al., 2017; Huey, 2007). The choice of reagent-ion determines the scope of the measurement in terms of ROC functionality and chemistry, while instrument construction and design impacts the range of detectable species in terms of volatility and reactivity (e.g., Riva et al., 2019; Krechmer et al., 2018). A range of reagent-ions are in common use and are selective toward specific ROC classes. Oxygenated, multi-functional organic gases can be detected as negative ions using iodide ($I^-/(H_2O)_n \cdot I^-$) (Lee et al., 2018, 2014), acetate ($CH_3O_2^-$) (Brophy and Farmer, 2015; Roberts et al., 2010), $CF_3O^-$ (Crounse et al., 2013, 2006), sulfur hexafluoride ($SF_6^-$) (Nah et al., 2018; Huey, 2007), nitrate ($NO_3^-$) and bromide ($Br^-$) (Rissanen et al., 2019; Bianchi et al., 2019) reagent-ions. Highly oxygenated organic species (with $C^* < 0.3 \times 10^6$ µg m$^{-3}$), together with low volatility inorganic species (e.g., $H_2SO_4$), can be detected with nitrate and bromide ionization at ambient pressure (Bianchi et al., 2019; Rissanen et al., 2019; Riva et al., 2019). $CF_3O^-$ effectively detects organic peroxides (Crounse et al., 2013, 2006), while $I^-$, $Br^-$, $NO_3^-$ and $SF_6^-$ detect a range of polar and acidic gases (Riva et al., 2019; Lee et al., 2018, 2014). Reduced VOCs, small oxygenated VOCs (e.g., methanol, ethanol, acetone, acetaldehyde) and reduced sulfur compounds (e.g., dimethyl sulfide and methanethiol) are readily detected as positive ions via proton transfer with hydronium ($H_3O^+$) reagent-ions (e.g., Kilgour et al., 2022; Pagonis et al., 2019; Krechmer et al., 2018; Yuan et al., 2017). $NO^+$ and $O_2^+$ allow detection of reduced VOCs, with proton affinities below that of water, that are generally not detectable with $H_3O^+$ (Jordan et al., 2009; Smith and Spanel, 2005). While $H_3O^+$ can detect functionalized VOCs, fragmentation is common and complicates the interpretation of mass spectra from complex samples (e.g., Coggon et al., 2024; Kilgour et al., 2024; Li et al., 2021; Pagonis et al., 2019; Yuan et al., 2017). To overcome some of the limitations induced by $H_3O^+$ ionization, fast separation techniques have been coupled to proton-transfer instruments (Claflin et al., 2021; Stockwell et al., 2021; Vermeuel et al., 2023b; Coggon et al., 2024; Kilgour et al., 2024), and other positive polarity reagent-ions have been applied to functionalized ROC. Benzene ($C_6H_6^+$) reagent-ions detect dimethyl sulfide, monoterpenes and sesquiterpenes with reduced fragmentation and higher selectivity compared to $H_3O^+$ (Kim et al., 2016). Water clusters (i.e., $(H_2O)_nH^+$) can detect a small subset of species detected by $H_3O^+$, such as dimethyl sulfide, with high selectivity (Blomquist et al., 2010). An array of oxygenated, multi-functional compounds in the intermediate to semi-volatile range can be detected using ammonium reagent-ions (e.g., Xu et al., 2022; Khare et al., 2022; Muller et al., 2020; Hansel et al., 2018), which provide some overlap in the fractions of ROC detected by negative polarity reagent-ions such as $I^-$ and $CF_3O^-$.

Compared to other positive polarity reagent-ions, ammonium ($NH_4^+$) adduction ionization is selective toward a wider range of multi-functional oxygenated compounds, such as carbonyls, alcohols, ethers, furans, and siloxanes (Xu et al., 2022; Khare

et al., 2022; Muller et al., 2020; Zhang et al., 2019; Zaytsev et al., 2019a; Berndt et al., 2019). Ambient observations have recently shown that $NH_4^+$ ionization can detect organic nitrates (Xu et al., 2022), while laboratory studies have demonstrated detection of organic peroxides (Zhou et al., 2018) and peroxy radicals (Hansel et al., 2018). Selected ion flow tube mass spectrometry (SIFT-MS) studies show that $NH_4^+$ ions form the strongest associations with carbonyl groups, relative to other oxygenates (e.g., alcohols and ethers) (Adams 2003). However, the conditions under which reagent-ions form and ion-molecule

reactions occur determine the dominant reagent-ion and ionization mechanism, which in turn controls the scope of detectable compounds and associated sensitivity. Possible reagent-ions include $NH_4 \cdot X_n^+$ (where X = $H_2O$ or $NH_3$ and n = 0, 1, 2, ...). In practice, multiple reagent-ions can be present, with $NH_4 \cdot H_2O^+$ providing optimal sensitivity to oxygenated compounds (Xu et al., 2022). Reactions with neutral analytes occur through ligand switching (Reaction R1), where the evaporation of X promotes softer adduct formation compared to $NH_4^+$ alone (i.e., n = 0) (Xu et al., 2022; Canaval et al., 2019; Adams et al.,

2003).

$$NH_4 \cdot X^+ + A \rightarrow A \cdot NH_4^+ + X \tag{R1}$$

Given an analyte, A, with larger $NH_4^+$ affinity (i.e., the negative enthalpy of the reaction: $NH_4^+ + A \rightarrow A \cdot NH_4^+$; e.g., Xu et al. (2022)) than X, the ionization reaction (R1) is exothermic (Adams et al., 2003). Therefore, $NH_4 \cdot X^+$ ligand switching reactions will proceed efficiently at or near the collision limit, with little importance of the reverse reaction unless reaction

timescales are long or the reaction is endothermic or only slightly exothermic (Xu et al., 2022; Zaytsev et al., 2019a). Reaction R1 is exothermic for the majority of oxygenates and multi-functional compounds (Xu et al., 2022; Adams et al., 2003; Canaval et al., 2019; Zaytsev et al., 2019a) (Table S3, Edward P. Hunter and Sharon G. Lias; Michael M. Meot-Ner (Mautner) and Sharon G. Lias); however, ion-molecule reactor (IMR) conditions must be selected to promote pure ion chemistry, optimize sensitivity, and minimize fragmentation.

Many CIMS reagent-ions provide access to complementary fractions of ambient ROC, and combining multiple reagent-ions in a single CIMS instrument can provide several advantages. First, reagent-ion switching can maintain the benefits of selectivity afforded by specific reagent-ions, while expanding the number of detectable compounds. For example, $H_3O^+$ and $NO^+/O_2^+$ are combined in proton-transfer reaction (PTR) and SIFT-MS instruments to expand detection capabilities to a broad range of alkanes, alkenes, aromatics, and some oxygenated species (e.g., Agarwal et al., 2014; Smith and Spanel, 2005). Further, the

combination of nitrate and bromide reagent-ions allows detection of a wide range of highly oxygenated molecules along with hydroperoxyl radicals, iodine compounds and sulfuric acid (He et al., 2023; Rissanen et al., 2019). Iodide, acetate and water cluster reagent-ions have been combined in laboratory studies, using repeated experiments rather than active reagent switching, to expand the range of detectable ROC (Aljawhary et al., 2013) and inform development of reaction mechanisms (e.g., Zhao et al., 2014a). Thus, combining reagent-ions can decrease the potentially large number of instruments required to characterize

a broad range of ROC classes (e.g., Heald and Kroll, 2020). Second, combining reagent-ions in a single instrument allows for direct comparison between the fractions of ROC detected by each chemical ionization reagent (e.g., Zaytsev et al., 2019a). For example, rapid switching between iodide and the acid-selective acetate reagent-ion informs detection of organic acids with iodide (e.g., Brophy and Farmer, 2015). Switching between $NH_4^+$ and $H_3O^+$ has benefits for measuring both reduced VOCs

and their early generation oxidation products (Zaytsev et al., 2019a, b), while also allowing a direct comparison between the subsets of ROC detected by each reagent-ion without the complications associated with differing instrument and inlet design (e.g., Riva et al., 2019).

Ambient atmospheric observations with $NH_4^+$ adduct ionization CIMS have focused primarily on urban environments, where $NH_4^+$ ion chemistry allows detection of oxygenated VOCs from volatile chemical products (Xu et al., 2022; Khare et al., 2022). $NH_4^+$/$H_3O^+$ reagent-ion switching has so far been limited to laboratory experiments demonstrating feasibility of switching (Muller et al., 2020) and application following laboratory oxidation of VOCs and oxygenated VOCs (Zaytsev et al., 2019a, b). We characterize $NH_4^+$/$H_3O^+$ reagent-ion switching using a Vocus Chemical Ionization Time-of-Flight Mass Spectrometer (Vocus-CI-ToFMS) using both laboratory standards and deployment at a rural forested site. We describe selection of ideal IMR conditions for $NH_4^+$/$H_3O^+$ reagent-ion switching, at the same temperature, with a focus on sensitivity, fragmentation, and prominence of competing ionization pathways. Using ambient reagent-ion switching data, we describe an approach to filter periods of impure ion chemistry, and once filtered, ambient observations allow us to directly compare the fractions of ambient ROC detected by $H_3O^+$ and $NH_4^+$. Our observations demonstrate that $NH_4^+$ is able to detect oxygenated and multi-functional biogenic ROC with both reduced fragmentation and higher selectivity compared to $H_3O^+$, illustrating a highly complimentary set of CIMS reagent-ions.

## 2   Methods and field site description

### 2.1   Instrument description

The Vocus-CI-ToFMS (Vocus-S, Tofwerk AG and Aerodyne Research Inc.) is described in detail elsewhere (Krechmer et al., 2018), with a brief description supplied here. Two features differentiate the Vocus-CI-ToFMS from other chemical ionization or proton transfer reaction time-of-flight mass spectrometers. First, the Vocus-CI-ToFMS is equipped with a focusing ion-molecule reactor (fIMR) which consists of a radio-frequency only quadrupole oriented around a 10 cm long resistive glass tube (Krechmer et al., 2018). The fIMR focuses ions toward the center-line, reducing ion losses to the walls and promoting ion transmission into a quadrupole high pass mass filter (BSQ). Second, polyether ether ketone (PEEK) tubing is used to establish flow restriction between ambient pressure and the fIMR. The use of PEEK at the instrument inlet reduces interactions between sampled air and more absorptive surfaces which impact transmission of S/IVOCs into the fIMR (Deming et al., 2019). These modifications to the Vocus-CI-ToFMS improve the ability to detect both reduced and oxidized ROC (Riva et al., 2019). The Vocus-CI-ToFMS used in this study has a mass resolving power of $\sim$5000 m dm$^{-1}$, a mass range of $\sim$50-500 m/z, with a 25 kHz ToF extraction frequency and is equipped with a multi-port reagent-ion injection, current-regulated discharge ion source. Details about instrument voltages are available in Table S1.

When using $H_3O^+$ ionization, we inject 20 cm$^3$ min$^{-1}$(STP) from the head-space above ultra high purity water (Millipore-Sigma, OmniSolv LC-MS) under vacuum into the discharge ion source. When switching to $NH_4^+$ ionization, we further add a flow from the head-space above a $\sim$1 w/w% solution of ammonium hydroxide (Oakwood Products Inc., Trace Metals Grade) in water to the ion source. Additionally, when switching between reagent-ions the voltages and pressure in the fIMR and the

ion optics are adjusted to compliment each reagent-ion, taking into account sensitivity, fragmentation, and purity of ionization chemistry; this is discussed in Sect. 3.1. A change between ionization modes results in hysteresis where the ion chemistry is impure. The filtering of hysteretic periods is discussed in Sect. 3.4.

## 2.2 Reactor pressure and voltage gradient

The fIMR collision energy can be controlled in part by adjusting the conditions that impact the velocity, free path, and thermal energy of ions; axial voltage gradient, pressure, and temperature. The temperature must remain constant during reagent-ion switching to allow for switching on 15-minute timescales. This restricts control of collisional energy to adjustments of the fIMR voltage gradient and pressure. To understand the impact that these parameters have on ion chemistry, we introduce a constant flow of dilute calibration standard while systematically changing the fIMR voltage gradient and pressure. We change the fIMR pressure in 0.1 mbar increments and hold it constant while we increase the fIMR front voltage by 10 V steps. With $NH_4^+$, we characterized from 2.5 to 3.5 mbar and from 45 to 65 $Vcm^{-1}$ (60-120 Townsends (Td)). For $H_3O^+$, we characterized from 1.5 to 2.5 mbar and from 45 to 65 $Vcm^{-1}$ (80-200 Td).

## 2.3 Sensitivity, detection limit, and fragmentation with standards

We calibrated 23 analytes from multi-component standardized gas cylinders (Apel-Riemer Environmental Inc.) to report sensitivities (counts $s^{-1}$ $ppt_v^{-1}$) and detection limits ($3\sigma$ of the background with 1s integration). The 23 analytes come from three separate multi-component cylinders where the composition was selected to avoid interferences from fragments (Table S2). Backgrounds were obtained using a zero air generator (Sabio Model 1001). We investigate fragmentation of molecular ions using single component samples of trans-2-hexen-1-ol (96.5 %, Acros Organics, Lot: A0340603), $\beta$-cyclocitral (92.3 %, ThermoFisher, Lot: 10237632), 2-hexenal (97.5 %, Oakwood Products Inc., Lot: 098868J07I), 2-hexanone (100 %, Oakwood Products Inc., Lot: 098350R22K), and 2-methyl-3-buten-2-ol (99.5 %, Oakwood Products Inc., Lot: 051281K14H). To calculate a molecular ion fraction, we average 15 s of 2 Hz data and fit peaks corresponding to molecular ions (i.e., $A \cdot H^+$ for $H_3O^+$ and $A \cdot NH_4^+$ for $NH_4^+$), identified fragments, and clusters then divide the molecular ion signal by the sum of all related peaks. fIMR conditions for $H_3O^+$ ionization during these experiments were 2.2 mbar, 60 °C, with a voltage gradient of 67.5 $Vcm^{-1}$ (140 Td) and a BSQ amplitude of 270 V. Using a 60 °C reaction chamber with $H_3O^+$ is lower than commonly reported in the literature ($\sim$80-100 °C) (e.g., Vermeuel et al., 2023a; Coggon et al., 2024); this choice arises from fIMR temperature constraints for $NH_4^+$ (Xu et al., 2022) and is discussed in more detail in Sect. 3.1. For $NH_4^+$ ionization the fIMR settings were 3.1 mbar, 60 °C, with a voltage gradient of 60 $Vcm^{-1}$ (90 Td) and a BSQ amplitude of 250 V. The BSQ frequency was 2.2 MHz, and the fIMR amplitude and frequency were 500V and 1.6 MHz respectively (Table S1).

## 2.4 Signal response to ambient relative humidity

To test the effect of relative humidity on sensitivity we varied the ratio of wet and dry flows (controlled with mass flow controllers, MKS Instruments Model 1179C Mass-Flo) to achieve a range of relative humidities. The relative humidity was mea-

sured inline (Omega Engineering Model HX71-V1). Downstream of the relative humidity measurement 10 cm$^3$ min$^{-1}$ (STP) of a certified gas standard (Apel-Riemer Environmental Inc. and Airgas for dimethyl sulfide) was added to the humidified flow

before being sampled to the Vocus-CI-ToFMS. Relative humidity ranged between 15 and 85 % during the experiments. Measurements alternated between elevated relative humidity and dry conditions, such that each measurement at elevated humidity could be directly compared to a dry (0 % RH) measurement immediately before.

## 2.5 Observations at Manitou experimental forest observatory

We deployed a Vocus-CI-ToFMS in Manitou Experimental Forest Observatory (MEFO) from September $3^{rd}$ to September

$24^{th}$ of 2021. MEFO is a rural ponderosa pine forest at middle elevation ($\sim$2,300 m) located $\sim$40 km northwest of Colorado Springs and $\sim$70 km southwest of Denver (39.1006°N, 105.0942°W). The ROC composition at this site is well characterized with emissions dominated by local biogenic sources (Hunter et al., 2017; Vermeuel et al., 2023a; Riches et al., 2024; Link et al., 2024). A full description of the field site can be found in Ortega et al. (2014). The Vocus-CI-ToFMS sample inlet was $\sim$4 m of 0.25 inch outer diameter perfluoroalkoxy (PFA) tubing situated $\sim$4 m above ground. The inlet flow was $\sim$3.8 L min$^{-1}$ (i.e.,

2.9 L min$^{-1}$ (STP)) pulled by a flow restricted bypass pump resulting in a laminar flow inlet (Reynolds number of $\sim$1150) corresponding to a residence time of $\sim$0.7 s. The inlet likely produced wall loss of oxygenated ROC and while the extent was not quantified, minimizing the inlet inner diameter and maximizing the flow rate, while maintaining laminar flow, serve to minimize inlet losses and tubing delays (Pagonis et al., 2017). The Vocus-CI-ToFMS sub-sampled 93-100 cm$^3$ min$^{-1}$ (i.e., 71 - 77 cm$^3$ min$^{-1}$ (STP)) perpendicular to the main inlet flow which helps prevent ambient aerosol clogging the capillary

inlet interface compared to a linear sub-sampling assembly (Jensen et al., 2023). We performed bi-hourly, 3 minute instrument backgrounds with ultra zero air (Airgas, UZA grade) followed by a 1 minute, single-point calibration with a certified calibrant mixture (Apel-Riemer Environmental Inc.) for both reagent-ions.

The Vocus-CI-ToFMS switched between NH$_4^+$ and H$_3$O$^+$ ionization on 15-minute time intervals during the deployment in MEFO. fIMR conditions for H$_3$O$^+$ ionization were 2.5 mbar, 60 °C, with a voltage gradient of 62 Vcm$^{-1}$ and a BSQ

amplitude of 350 V; and for NH$_4^+$ ionization were 3.1 mbar, 60 °C, with a voltage gradient of 65 Vcm$^{-1}$ and a BSQ amplitude of 250 V. The fIMR settings correspond to $E/N$ values of 114 Td and 96 Td for H$_3$O$^+$ and NH$_4^+$ respectively. The BSQ frequency was 2.2 MHz, and the fIMR amplitude and frequency were 450 V and 1.3 MHz respectively (Table S1). fIMR parameters for both reagent-ions were informed by experiments detailed in Sect. 3.1.

## 2.6 ARTofMELT Expedition on icebreaker (I/B) *Oden*

We deployed the Vocus-CI-ToFMS on-board Swedish I/B *Oden* from May $7^{th}$ to June $15^{th}$ of 2023 as part of the Atmospheric Rivers and the onseT of Arctic MELT measurement expedition (ARTofMELT). The cruise on I/B *Oden* took place within the pack ice and marginal ice zone between Svalbard and Greenland in the Fram Strait. The Vocus-CI-ToFMS switched between NH$_4^+$ and H$_3$O$^+$ ionization on 15-minute time intervals during the expedition. The Vocus-CI-ToFMS was mounted to the floor of of a sea container on I/B Oden's 4th deck using metal plates and five high deflection vibration isolation feet (Barry Controls

Model 2K2-BA-90) and was mounted to an open wall using two wire rope isolators (Enidine Model WR6-850-10-E).

Ambient air was sampled through a 0.375 inch outer diameter PFA tube ~15 m in length. The entire length of tubing was insulated and heated to 30 °C in three separately controlled sections. The Vocus-CI-ToFMS inlet flow was driven by a vacuum pump (Agilent IDP-7 Dry Scroll) regulated by a mass flow controller (15 L min$^{-1}$(STP)). A deuterated internal standard (containing dimethylsulfide-d$_3$, acetone-d$_6$, 2-hexanone-d$_4$, and mesitylene-d$_{12}$) was injected near the top of the Vocus-CI-ToFMS inlet through a 0.125 inch outer diameter PFA tube at a rate of 5 cm$^3$ min$^{-1}$(STP) into the total inlet flow of 15 L min$^{-1}$(STP) yielding ~333 ppt$_v$ from the nominally 1 ppm standardized cylinder (Apel-Riemer Environmental Inc.). The inlet assembly was affixed to a metal pipe extending from the top of the sea container toward the bow of the ship at a ~40° angle ~3 m above the top of the container or 25 m above the water/ice surface. The use of internal standards during this expedition allows us to analyze reagent-ion hysteresis (Sect. 3.4) using a persistent and known set of ions. fIMR conditions for H$_3$O$^+$ ionization during ARTofMELT were 2.2 mbar, 60 °C, a voltage gradient of 67.5 Vcm$^{-1}$ (140 Td) and a 270 V BSQ amplitude. For NH$_4^+$ ionization the fIMR settings were 3.1 mbar, 60 °C, a 60 Vcm$^{-1}$ (90 Td) voltage gradient, and a 250 V BSQ amplitude. The BSQ frequency was 2.2 MHz, and the fIMR amplitude and frequency were 500V and 1.6 MHz respectively (Table S1) We reserve further analysis of the ARTofMELT data set for future work.

## 2.7 Data analysis

Raw mass spectral data were collected with Acquility (version 2.3.18) and ToFDAQ (version 1.99) (Tofwerk AG) and processed in Tofware (version 3.2.5, Tofwerk AG and Aerodyne Research Inc.). For MEFO data, the time resolution was pre-averaged from 1 Hz to 0.1 Hz. For ARTofMELT data, the time resolution was pre-averaged from 2Hz to 1Hz. All data was mass calibrated, baseline subtracted, and peak fit in Tofware. Time-integrated high resolution ion signals were exported for further analysis in python (version 3.9.12). Responses in the $E/N$ scans in Sect. 3.1 were interpolated using a linear interpolation on a triangular grid (using matplotlib.tri.LinearTriInterpolator). $C^*$ values are estimated using EPI Suite (US EPA).

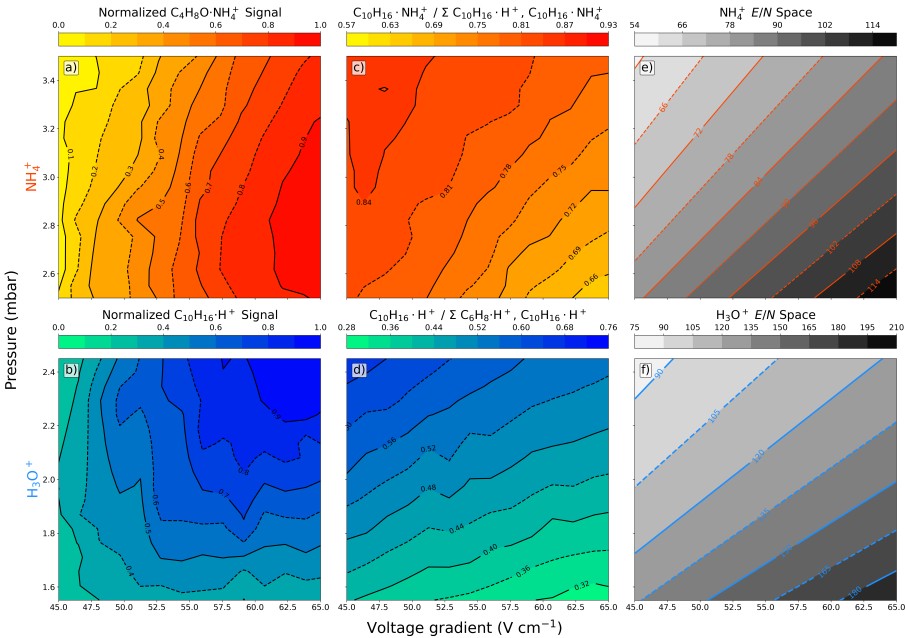

**Figure 1.** (a-d) Contour plots of fIMR pressure and voltage gradient scans with a constant concentration of analyte ($10\ \text{ppb}_\text{v}$) introduced into the Vocus-CI-ToFMS. Normalized signal intensity for (a) methyl ethyl ketone measured with $NH_4^+$ ($C_4H_8O \cdot NH_4^+$) and (b) $\alpha$-pinene measured with $H_3O^+$ ($C_{10}H_{16} \cdot H^+$). (c) Fractional contribution of the $NH_4^+$ molecular ion ($C_{10}H_{16} \cdot NH_4^+$) to the total $\alpha$-pinene signal (i.e., the sum of the proton transfer product ($C_{10}H_{16} \cdot H^+$) and the molecular ion). (d) Fractional contribution of the $H_3O^+$ molecular ion ($C_{10}H_{16} \cdot H^+$) to the total $\alpha$-pinene signal (i.e. the sum of the $\alpha$-pinene fragment ($C_6H_8 \cdot H^+$) and the molecular ion). Contour plots of calculated $E/N$ values over the scanned space for both (e) $NH_4^+$ and (f) $H_3O^+$.

## 3 Results and discussion

### 3.1 Ion-molecule reactor pressure and voltage gradient

$H_3O^+$ and $NH_4^+$ ionization operate optimally at differing combinations of fIMR pressure, voltage gradient, and temperature (Xu et al., 2022; Gouw and Warneke, 2007), all of which impact the reduced electric field ($E/N$) of the fIMR:

$$E/N = \frac{T \times \Delta V \times k_B}{l_{imr} \times p} \tag{1}$$

Where $T$ is the temperature (kelvin), $\Delta V$ is the difference between the front and back voltage (volts), $l_{imr}$ is the fIMR length (meters), $k_B$ is the Boltzmann constant (joules per kelvin), and $p$ is pressure (pascals or joules per cubic meter). $E/N$ has units of townsends (Td, 1 Td $= 1 \times 10^{-17}$ Vcm$^2$) and describes ion velocity and collisional energy. High $E/N$ values promote increased fragmentation and reduced clustering while low $E/N$ values promote cluster formation and reduced fragmentation. We analyzed relative sensitivity, fragmentation, and prevalence of ionization pathways, while varying the fIMR pressure and voltage gradient with a constant temperature of $60°C$ (Fig. 1) to inform our selection of fIMR settings. Because fIMR temperature takes tens of minutes to stabilize, we selected a constant $60\,°C$ fIMR temperature to promote $NH_4^+ \cdot H_2O$ clusters in $NH_4^+$ ionization. This is lower than most $H_3O^+$ fIMR temperatures ($\sim$80-100 $°C$), however, the effect of lower fIMR temperature on the reagent-ion distribution can be mitigated by adjusting other fIMR settings.

Selecting $H_3O^+$ ionization fIMR parameters requires balancing fragmentation and sensitivity. We observe a large increase ($>$60 % at 2.4 mbar) in the sensitivity to the molecular ion ($C_{10}H_{16} \cdot H^+$) with increased fIMR voltage gradient (Fig. 1b) which arises from three main factors. First, reduced residence time with increased voltage gradient (163 µs or 113 µs at 45 Vcm$^{-1}$ or 65 Vcm$^{-1}$ respectively with pressure and temperature of 2.4 mbar and 333.15 K) could increase ion transmission through the fIMR. Second, at low $E/N$ protonated water clusters contribute to the ionization of $\alpha$-pinene. The production of protonated water clusters is evident from the reduced benzene sensitivity at lower $E/N$ (Fig. A1) (Gouw and Warneke, 2007). Water clusters have a higher proton affinity (i.e., the negative enthalpy of the reaction: $H^+ + A \rightarrow A \cdot H^+$) compared to water (Hunter and Lias, 1998), and $\alpha$-pinene has a higher proton affinity than both water and the first water cluster (i.e., $(H_2O)_2$), making ionization reactions with both exothermic. In contrast, benzene has a proton affinity higher than water but lower than the first cluster, making the ionization reaction of benzene with $(H_2O)_2H^+$ endothermic and unlikely. Therefore the formation of water clusters will reduce the sensitivity to benzene while increasing the sensitivity to $\alpha$-pinene. Third, the known $H_3O^+$ $\alpha$-pinene fragment, $C_6H_8 \cdot H^+$ has a larger contribution to the total $\alpha$-pinene signal at higher $E/N$ (Fig. 1d) which suggests that part of the increased sensitivity to $C_{10}H_{16} \cdot H^+$ at higher pressures is also attributable to reduced loss to fragmentation. The combined responses of transmission, fragmentation, and proton affinity to changes in voltage and pressure with $H_3O^+$ ionization result in a sensitivity that does not follow $E/N$ directly (Fig. 1b & 1f).

Selecting $NH_4^+$ ionization fIMR parameters requires balancing between signal intensity and purity of ion chemistry. Similar to $H_3O^+$, with $NH_4^+$ we observe increased signal intensity of the methyl ethyl ketone molecular ion ($C_4H_8O \cdot NH_4^+$) at higher voltage gradients (Fig. 1a). In contrast to $\alpha$-pinene detection with $H_3O^+$, sensitivity to $C_4H_8O \cdot NH_4^+$ is not impacted by fragmentation and changing reagent-ion proton affinity across the range in voltage gradient. The sensitivity to the $C_4H_8O \cdot NH_4^+$

ion is highest at high $E/N$, but the change in sensitivity is mostly dependent on the voltage gradient and is less impacted

by the fIMR pressure (Fig. 1a). We observe a similar trend for other oxygenated ROC species (Fig. A2). The vertical contours in the $NH_4^+$ sensitivity contrasted with the L-shaped contouring in the $H_3O^+$ sensitivity (Fig. 1a & 1b) supports the three-factor dependence for $H_3O^+$ on transmission, fragmentation, and proton affinity, and points to transmission as the major factor impacting sensitivity with $NH_4^+$ ionization. However, at high $E/N$ we observe impurities in the ionization chemistry (i.e., proton transfer products ($A \cdot H^+$) occurring for $\alpha$-pinene) with $NH_4^+$ ionization (Fig. 1c). Proton transfer ionization under

$NH_4^+$ is undesirable because it provides lower selectivity, leads to higher fragmentation rates compared to the ligand-switching mechanism, and complicates interpretation of the mass spectrum. $\alpha$-pinene has a lower ammonium affinity than $H_2O$ (Canaval et al., 2019) which makes the ligand switching reaction (R1) endothermic and thus dependent on increased collisional energy at higher voltage gradients (Xu et al., 2022). We observe a larger contribution of the proton transfer product at higher $E/N$, which is consistent with electric field-induced production of $C_{10}H_{16} \cdot H^+$ through internal proton transfer (Xu et al., 2022).

Alternatively, it is also possible that $C_{10}H_{16} \cdot H^+$ production is enhanced by declustering of the $NH_4^+ \cdot H_2O$ ions to form $NH_4^+$ which is more likely to undergo proton transfer reactions directly, without the need for internal proton transfer, due to the lower proton affinity of $NH_3$ compared to $\alpha$-pinene (Canaval et al., 2019). Regardless of the mechanism, formation of proton transfer products is ideally avoided and we find that their production is minimized at low $E/N$ (Fig. 1c). We note that the formation of secondary clusters (i.e., $NH_4 \cdot H_2O \cdot A^+$ and $NH_4 \cdot NH_3 \cdot A^+$) is negligible over the entire investigated $E/N$ space (Fig. S1 &

S2). The sensitivity to methyl ethyl ketone, and prevalence of undesirable reaction products ($C_{10}H_{16} \cdot H^+$) are optimal under opposing conditions in the voltage gradient-pressure space, such that optimal fIMR parameters for $NH_4^+$ require a compromise between sensitivity and purity of ionization chemistry.

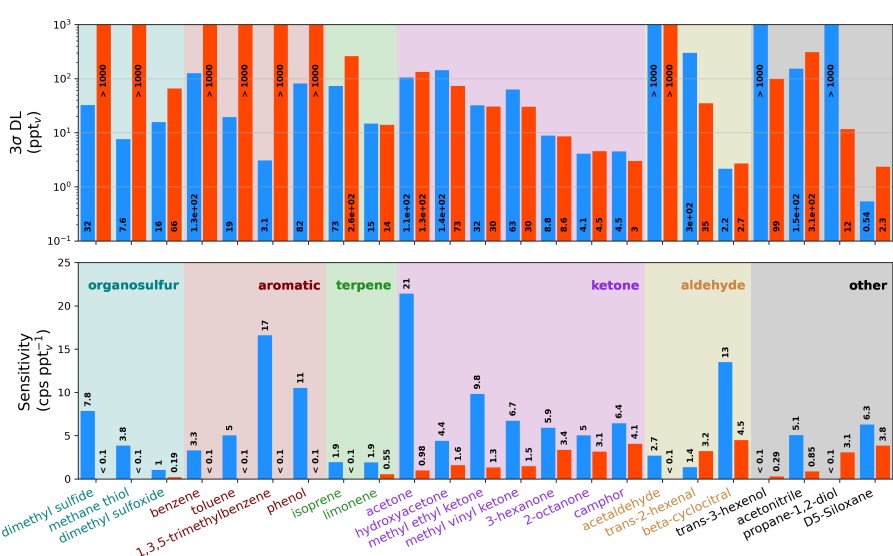

**Figure 2.** Sensitivities (bottom) and detection limits (top, log y-axis) for the Vocus-CI-ToFMS with $NH_4^+$ ionization (orange) and $H_3O^+$ ionization (blue) for 23 analytes from standardized gas cylinders, grouped by functional group/compound type. Sensitivities and detection limits (DLs) are calculated for the molecular ion only (i.e., $A \cdot NH_4^+$ for $NH_4^+$, or $A \cdot H^+$ for $H_3O^+$), and mass spectral fragments are not included. DL are calculated as $3\sigma$ over a 600 second background at 1 Hz.

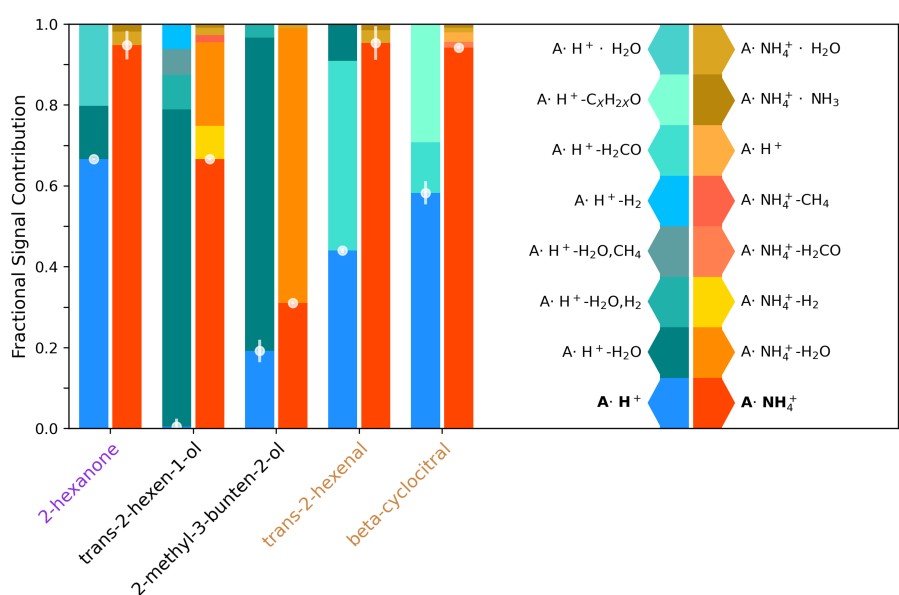

**Figure 3.** Molecular ion fraction and the contribution of various non-molecular ions for 5 analytes using $H_3O^+$ ionization (blue/left) and $NH_4^+$ ionization (orange/right). The contribution of the molecular ion is highlighted with a scatter plot and uncertainty bars are standard deviation across 30 mass spectra at 2 Hz. Data represented in this figure is shown in Table S6.

## 3.2 Sensitivities, detection limits, and fragmentation

Direct calibrations demonstrate the selectivity of $NH_4^+$ relative to $H_3O^+$ ionization for a range of carbonyls, hydrocarbons, alcohols, and organic sulfur compounds (Fig. 2). We observe that $H_3O^+$ is capable of detecting nearly every compound in this set of analytes from standardized gas cylinders at the molecular ion, which demonstrates the utility of $H_3O^+$ as a general reagent-ion that allows for detection of reduced and some oxidized species. In contrast, $NH_4^+$ ionization is more selective toward oxygenates, including saturated and unsaturated ketones, unsaturated aldehydes, and the multi-functional propane-1,2-diol. $NH_4^+$'s selectivity toward oxygenates demonstrates its utility for expanding the range of compounds detectable with a single instrument, and supporting identification of molecular ions and fragments detected simultaneously with $H_3O^+$. In addition, $NH_4^+$ does not detect aromatics, small alkenes, and reduced sulfur compounds that $H_3O^+$ detects well, demonstrating the complimentary nature of these reagent-ions. While Fig. 2 suggests that $H_3O^+$ detects dimethyl sulfoxide (DMSO), an oxidation product of dimethyl sulfide (Barnes et al., 2006), with a detection limit (DL) of 16 $ppt_v$, this DL is optimistic because DMSO peak separation is hindered by isobaric ions of protonated benzene and a protonated water cluster of acetic acid. We therefore expect that DMSO cannot be detected with $H_3O^+$ in the Vocus-CI-ToFMS at concentrations relevant to the marine boundary layer (i.e., <100 $ppt_v$, Putaud et al. (1999); Sciare et al. (2000); Legrand et al. (2001); Nowak et al. (2001)).

For the compounds detected with both ionization modes, sensitivities and detection limits for $H_3O^+$ and $NH_4^+$ are in the same order of magnitude (Fig. 2 & A3). $NH_4^+$ detects the subset of ketones and the unsaturated aldehydes shown in Fig. 2 with a lower or similar DL to $H_3O^+$. Propane-1,2-diol, trans-3-hexenol, and D5-siloxane suggest that $NH_4^+$ has a greater ability than $H_3O^+$ to detect oxygenated and functionalized compounds, but this is not broadly apparent across the families of compounds we calibrated directly (Fig. 2). This likely arises because the analytes shown in Fig. 2 are limited to compounds amendable to gas cylinder calibration, and are therefore biased toward VOCs and S/IVOCs with minimal oxygenation and relatively high volatility (i.e., $C^* > 9 \times 10^4$ µg m$^{-3}$; Table S4). Despite the compromises in the fIMR temperature made to allow for the switching system (Sect. 3.1) the sensitivities for $NH_4^+$ ionization reported here are similar to the sensitivities reported in recent $NH_4^+$ literature (Khare et al., 2022; Xu et al., 2022) (Table S5).

Compared to $H_3O^+$ ionization, $NH_4^+$ ionization reduces molecular ion fragmentation for functionalized compounds (Fig. 3). We use a molecular ion fraction (Fig. 3; ratio of the molecular ion signal to the total signal from the molecular ion, fragments, and clusters) to analyze the contributions of molecular ions and mass spectral fragments from both $NH_4^+$ and $H_3O^+$ ionization for a series of analytes complimentary to those calibrated with standardized gas cylinders (Fig. 2). Alcohols fragment substantially using both reagent-ions; with $H_3O^+$, trans-2-hexenol fragments almost completely away from the molecular ion resulting in a near zero molecular ion fraction. This is consistent with the negligible sensitivity to the similarly structured trans-3-hexenol molecular ion (Fig. 2) with $H_3O^+$ owing to fragmentation (e.g., Pagonis et al., 2019). In contrast, $NH_4^+$ ionization detects trans-2-hexenol with a molecular ion fraction of 0.67. For the tertiary alcohol, 2-methyl-3-buten-2-ol, (2,3,2-MBO), we observe substantial fragmentation with both ionization modes, but a higher molecular ion fraction under $NH_4^+$ (i.e., 0.31 with $NH_4^+$ and 0.19 with $H_3O^+$). The ketone and aldehydes sampled only fragmented substantially under $H_3O^+$ ionization, while $NH_4^+$ retains the molecular structure leading to high molecular ion fraction. This is consistent with the lower $NH_4^+$ detection

limit for the majority of ketones we examined (with the exceptions of acetone and 2-octanone) and the aldehyde trans-2-hexenal (Fig. 2). Our observations suggest that reduced fragmentation has a larger impact on detection capability of the two reagent-ions for more highly oxidized compounds with multiple functional groups. This is observed for propane-1,2-diol which is

readily detected with $NH_4^+$ but not with $H_3O^+$; the detection of oxidized ROC is discussed further in Sect. 3.5. Overall, these observations demonstrate the high selectivity of $NH_4^+$ ionization for oxygenates and the benefits of reduced fragmentation with $NH_4^+$. Coupling the detection of reduced ROC and organic sulfur from $H_3O^+$ with the detection of oxygenates from $NH_4^+$ expands the fraction of atmospheric ROC that we can detect with a single instrument.

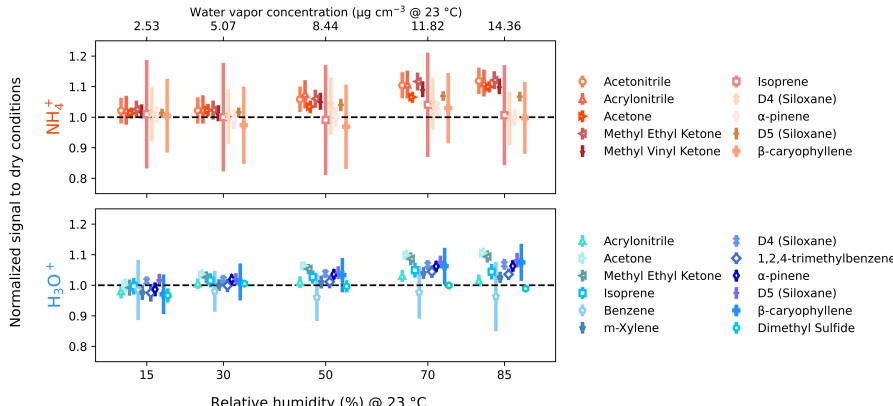

**Figure 4.** Signal dependence on sample relative humidity for $NH_4^+$ (top, orange), and $H_3O^+$ (bottom, blue). Measurements were made at relative humidities of 15, 30, 50, 70, and 85 %; points are offset from these values for visibility. Ethanol, benzene, m-xylene, and 1,2,4-trimethylbenzene are excluded for $NH_4^+$ due to low signal. Acetonitrile and ethanol are omitted for $H_3O^+$ due to low transmission through the BSQ. Methyl ethyl ketone is omitted for $H_3O^+$ due to interference of the reagent-ion cluster $(H_2O)_3 \cdot H^+$. The error bars represent propagated relative deviations in dry and humidified signals.

### 3.3 Impact of sample relative humidity

Previous studies have characterized the significant humidity dependence of sensitivity in various CIMS instruments to under-stand and correct for changing ambient humidity (e.g., Warneke et al., 2001; Gouw and Warneke, 2007; Kari et al., 2018; Zaytsev et al., 2019a). Humidity-driven changes in reagent-ion chemistry, and therefore sensitivity, are generally small in the Vocus-CI-ToFMS due to the large flow of water vapor (i.e., $20\,\mathrm{cm^3\,min^{-1}}$ (STP)) injected into the ion source (Krechmer et al., 2018; Khare et al., 2022). Varying sample humidity with constant analyte concentration demonstrates low humidity dependence

with both $NH_4^+$ and $H_3O^+$ ionization across a range of reduced and oxygenated ROC (Fig. 4). We note an approximately 10 % increase in the $NH_4^+$ sensitivity to nitriles and oxygenates while alkene sensitivities remain unchanged up to 85 % RH. We also observe a slight (5-10%) increase in sensitivity with humidity for oxygenated species with $H_3O^+$, while alkene sensitivities are less affected. The low humidity dependence of the Vocus-CI-ToFMS has been demonstrated previously for $H_3O^+$ for a variety of analytes (Krechmer et al., 2018; Kilgour et al., 2022; Li et al., 2024) and for a select number of small oxygenates, alkenes,

and acetonitrile with $NH_4^+$ (Khare et al., 2022; Xu et al., 2022). We demonstrate the low dependence of sensitivity on sample humidity with $NH_4^+$ ionization under different instrumental conditions and for a selection of analytes including oxygenated alkenes and siloxanes (Fig. 4).

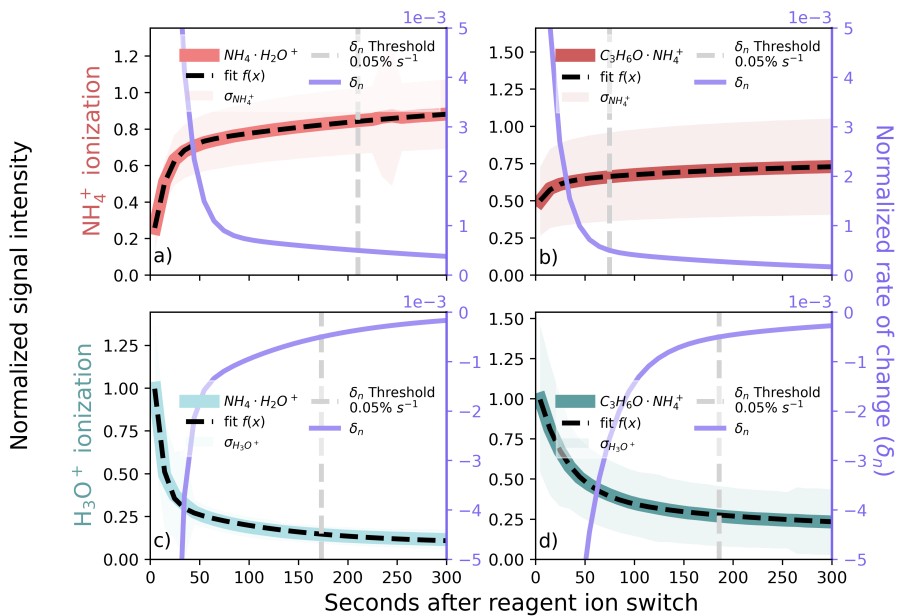

**Figure 5.** Ion signal after a reagent-ion switch for $NH_4^+$ (a & b) and $H_3O^+$ (c & d) in the MEFO data, showing $NH_4 \cdot H_2O^+$ ions (a & c) and $C_3H_6O \cdot NH_4^+$ (b & d). We grouped ion signals by the time after a switch and normalized the mean of each group by the maximum, and normalized means were fit with a bi-exponential function. The derivative of the fit ($\delta_n$) is displayed on the right axes (purple traces) and is used as a metric to filter reagent-ion hysteresis. A summary of the amount of data removed as a function of the selected threshold for these ions is shown in Fig. S4.

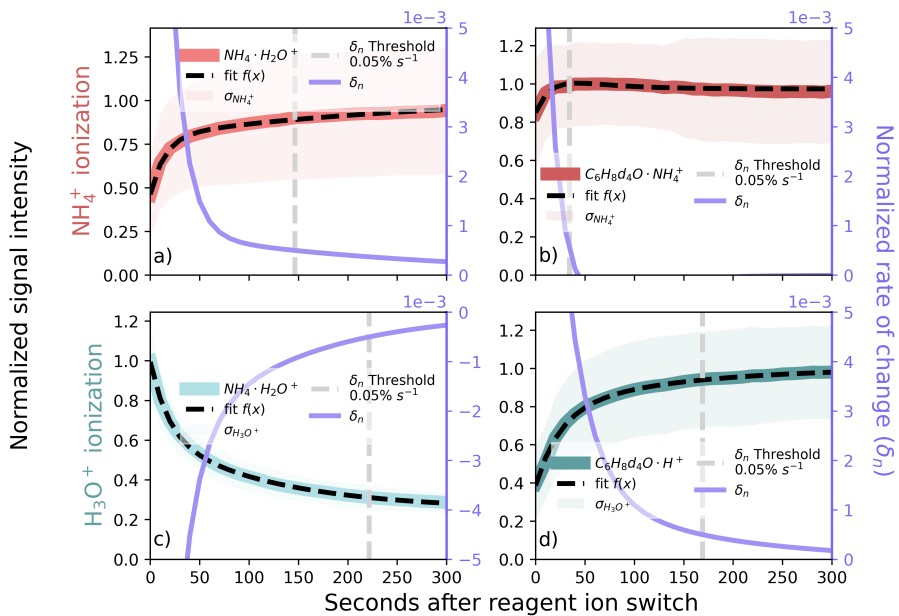

**Figure 6.** Ion signal after a reagent-ion switch for $NH_4^+$ (a & b) and $H_3O^+$ (c & d) in the ARTofMELT data, showing $NH_4 \cdot H_2O^+$ ions (a & c), $C_6H_8d_4O \cdot NH_4^+$ (b) and $C_6H_8d_4O \cdot H^+$ (d) internal standard ions. We grouped ion signals by the time after a switch and normalized the mean of each group by the maximum, and normalized means were fit with a bi-exponential function. The derivative of the fit ($\delta_n$) is displayed on the right axes (purple traces) and is used as a metric to filter reagent-ion hysteresis. A summary of the amount of data removed as a function of the selected threshold for these ions is shown in Fig. S5.

### 3.4 Removal of reagent-ion hysteresis from switching

Reagent-ion chemistry does not stabilize immediately upon switching between $NH_4^+$ and $H_3O^+$ ionization. This reagent-ion
switch requires adjustment of instrument conditions that impact ion chemistry; these include the reagents introduced into
the ion source, the fIMR pressure and voltage gradient, and downstream ion optic voltages (Table S1). However, instrument
conditions for each reagent-ion (Sect. 3.1) are such that analyte detection through the alternate ionization pathway is possible
with both $NH_4^+$ and $H_3O^+$ ionization (e.g., Zaytsev et al., 2019a). This is in contrast to some other reagent-ion pairs (e.g.,
$CH_3O_2^-$ and $I^-$) where instrument conditions differ drastically, and so hysteresis is not observed (e.g., Brophy and Farmer,
2015). As a result, when $NH_4^+$ and $H_3O^+$ are paired in a single instrument we observe a distinct transitional period of reagent-
ion hysteresis following each reagent-ion switch. The hysteretic period arises from: (1) changes in ion transmission due to
instrument conditions, such as ion optics and fIMR settings, which are fast (ones to tens of seconds), and (2) changes in the
reagent-ion speciation due to the presence or absence of $NH_3(g)$, which is slower (tens of seconds to minutes). Periods of
hysteresis must be characterized and removed to ensure stable and consistent measurements without drifting sensitivities over
the course of each 15-minute measurement period. We accomplish this by monitoring ion stability over a large number of
repeated switches.

We compare the utility of three ion-types as markers to quantify the timescale of reagent-ion switching hysteresis: $NH_4^+$
reagent-ion signal, a persistent ambient $NH_4^+$-adduct ion; and known and persistent $NH_4^+$-adduct or proton-transfer molecular
ions from an internal standard infused in the sampling inlet. We selected ions primarily measured with $NH_4^+$ ionization because
the influence of $NH_3(g)$ is observed under both $H_3O^+$ and $NH_4^+$ ionization modes, whereas the influence of $H_2O(g)$ reagent-
ions are not observed in $NH_4^+$ mode (Fig. S3). This arises because $NH_3(g)$ has a higher proton affinity than $H_2O(g)$ (Hunter
and Lias, 1998) which causes any $NH_3(g)$ present in the ion source and fIMR to readily form $NH_4^+$ or $NH_4 \cdot H_2O^+$ at the
expense of $H_3O^+$ formation.

### 3.4.1 Filtering hysteresis with reagent and persistent ambient ions

In the absence of a known and persistent signal from an internal standard to diagnose reagent-ion hysteresis, we compare the use
of $NH_4 \cdot H_2O^+$ (i.e., the prominent ammonium reagent-ion signal) and $C_3H_6O \cdot NH_4^+$ (i.e., a ubiquitous analyte-ammonium
molecular ion) under both ionization methods. Both $NH_4 \cdot H_2O^+$ and $C_3H_6O \cdot NH_4^+$ decay after switching to $H_3O^+$ ionization
(Fig. 5c, d; $NH_3(g)$ depletion in the fIMR) and intensify after switching to $NH_4^+$ ionization (Fig. 5a, b; $NH_3(g)$ accumulation
in the fIMR). We grouped a total of 558 hours of ambient $NH_4^+/H_3O^+$ 15-minute reagent-ion switching observations from
MEFO (Sect. 2.5) by time after a reagent-ion switch. We normalized the mean signal (in 10 second intervals starting at 5
seconds into a switch) to the maximum and fit the normalized data with a bi-exponential function (Fig. 5). The bi-exponential
function describes changes in both instrument conditions (fast) and equilibration of $NH_3(g)$ in the ion source and fIMR that
drives reagent-ion chemical speciation (slow). We use the derivative of this decay function ($\delta_n$) to quantify a normalized rate
of change in the ion signal as a function of the time after a switch (Purple lines in Fig. 5a-d). We use $\delta_n$ to set a threshold
for filtering hysteresis, removing data before $\delta_n$ reaches the set threshold. For all ions in Fig. 5, $\delta_n$ changes rapidly in the first

$\sim100$ s after a reagent-ion switch and slowly approaches but does not reach zero on the measurement timescale (900 s) (Fig. S4), likely due to the time scale for complete $NH_3(g)$ equilibration with instrument surfaces.

Monitoring $NH_4 \cdot H_2O^+$ $\delta_n$ has the benefit of being directly related to the abundance of reagent-ion; however, both the decay of $NH_4 \cdot H_2O^+$ in $H_3O^+$ ionization mode and its initial increase in $NH_4^+$ ionization mode is driven largely by changes in the BSQ mass range. We are able to avoid the impacts of changing BSQ mass range by monitoring an ion with higher $m/z$. Additionally, using an analyte for filtering reagent-ion hysteresis means that we are using a direct measurement of the formation of analyte ions for $NH_4^+$ ionization and a direct measurement of contamination from other reagent-ion chemistry with $H_3O^+$ ionization. Therefore, in the absence of an internal standard (Sect. 3.4.2), we use the persistent ambient ion $C_3H_6O \cdot NH_4^+$ to monitor hysteresis. This approach brings two major complications: (1) variable contributions of isomers with potentially disparate sensitivities (e.g., acetone and propionaldehyde), and (2) potentially variable ambient concentrations. A switch-by-switch analysis of hysteresis from MEFO (available as Fig. S6) shows that the 0.05 % s$^{-1}$ $\delta_n$ cutoffs for $C_3H_6O \cdot NH_4^+$ calculated in Fig. 5b, d do not capture the majority of the switch-by-switch cutoffs (37 % for $NH_4^+$ and 39 % for $H_3O^+$). Therefore, if a persistent ambient ion is used to diagnose hysteresis timescales, this should be done on a switch-by-switch basis. This variability may be associated with ambient variations in the $C_3H_6O \cdot NH_4^+$ signal which can be avoided by applying our method described in Fig. 5 to a persistent and known signal from an internal standard (Sect. 3.4.2).

The choice of $\delta_n$ threshold represents a compromise between ion chemistry stability and data loss. For both reagent and analyte ions, the amount of data removed becomes very sensitive to a small decrease in the $\delta_n$ threshold below $\sim0.05$ % s$^{-1}$ (Fig. S4). During the deployment in MEFO, a 0.05 % s$^{-1}$ threshold applied to $C_3H_6O \cdot NH_4^+$ results in the loss of $\sim185$ s (Fig. 5d) and $\sim75$ s (Fig. 5b) of data per switch with $H_3O^+$ and $NH_4^+$ ionization, respectively. Optimizing ion chemistry stability, while preserving data coverage results in the loss of $\sim 260$ s (75 s for $NH_4^+$ and 185 s for $H_3O^+$) of data on a reagent switching full cycle (1800 s), corresponding to $\sim 86$ % data retention for 15-minute switching intervals.

### 3.4.2 Filtering hysteresis with reagent and internal standard ions

When an internal standard signal is available, as in the ARTofMELT expedition (Sect. 2.6), reagent-ion hysteresis can be more reliably monitored using known unique and persistent molecular ions. We applied the $\delta_n$ thresholding method to a 2-week period from the ARTofMELT data set (from May 17$^{th}$ to May 31$^{st}$ of 2023). We quantify the timescale of reagent-ion hystersis by monitoring the internal standard signal of 2-hexanone-d$_4$ as $C_6H_8d_4O \cdot NH_4^+$ with $NH_4^+$ ionization (Fig. 6b) and as $C_6H_8d_4O \cdot H^+$ with $H_3O^+$ ionization (Fig. 6d). For direct comparison to Sect. 3.4.1, we also use the reagent ion $NH_4 \cdot H_2O^+$ under both $NH_4^+$ (Fig. 6a) and $H_3O^+$ (Fig. 6c). Notably, in this marine environment the ambient $C_3H_6O \cdot NH_4^+$ signal is highly variable, precluding its use for filtering reagent-ion hysteresis (Fig. S7), further motivating the use of an internal standard to diagnose reagent hysteresis. A 0.05 % s$^{-1}$ threshold applied to 2-hexanone-d$_4$ results in the loss of 34 seconds from $NH_4^+$ ionization after a switch (Fig. 6b) and 168 seconds from $H_3O^+$ ionization (Fig. 6d). This results in $\sim19$ % of $H_3O^+$ data being removed and $\sim4$ % of $NH_4^+$ ionization from 15-minute switching, or $\sim89$ % total data retention over a full (1800 s) switching cycle. It's worth noting that under both ionization modes, the hysteresis timescale for the $NH_4 \cdot H_2O^+$ is longer (Fig. 6a & c, S8) than the 2-hexanone-d$_4$ internal standard ion. This suggests that a conservative approach to monitoring ion chemistry

could be to use the reagent-ions for establishing hysteresis timescales. However, the impacts of the BSQ on reagent-ions raises concerns of how representative the reagent-ion signal is of true composition in the fIMR (Krechmer et al., 2018; Khare et al., 2022; Xu et al., 2022). A switch-by-switch analysis of variation in the hysteresis timescale (available as Fig. S8) demonstrates that the cutoffs calculated in Fig. 6 for 2-hexanone-d$_4$ capture the majority of variability in hysteresis timescale for independent switches (75 % for $NH_4^+$ and 72 % for $H_3O^+$) .

Our results using an internal standard for reagent-ion hysteresis filtering are qualitatively similar to the analysis above (Sect. 3.4.1) using a persistent ambient ion, though there is less data removed from $NH_4^+$ ionization. Rather than resulting from use of an ambient or internal standard ion, this difference in hysteresis time is likely the result of having more similar BSQ settings between the two ionization modes during the ARTofMELT campaign (Table S1). The smaller change in BSQ settings results in a faster change in instrument conditions impacting ion transmission. The approach we describe here can be applied

easily to other instruments and at different instrument conditions (e.g., fIMR temperature and pressure) to best balance the need for both measurement stability and data coverage. While the use of an internal standard signal is ideal for diagnosing reagent-ion switching hystersis, the choice of both product molecular ions and rate-of-change threshold must be optimized for each application and sampling environment. Overall, our results are qualitatively similar to hysteresis timescales suggested by Zaytsev et al. (2019a) for $NH_4^+$ and $H_3O^+$ switching in an Ionicon PTR3: $\sim$120 s and $\sim$60 s of data removed after switching

to $H_3O^+$ and $NH_4^+$ ionization, respectively.

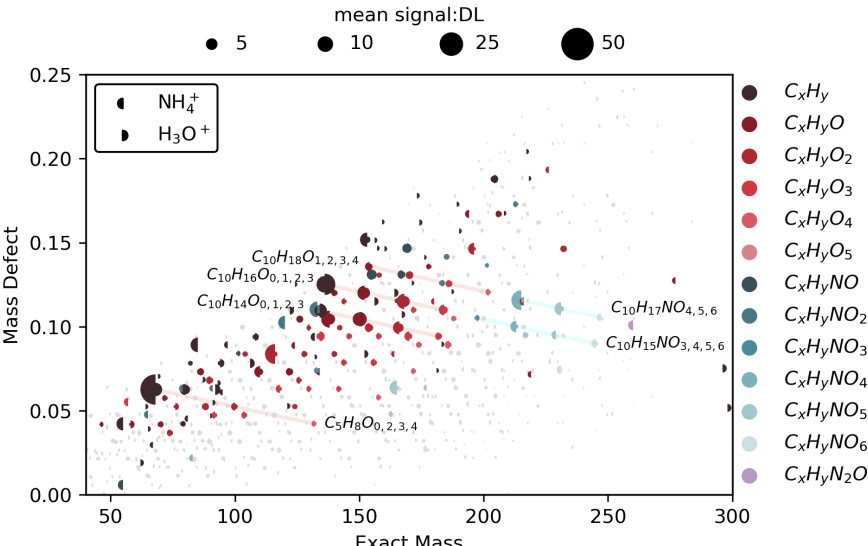

**Figure 7.** Campaign mean mass defect plots for $NH_4^+$ and $H_3O^+$ ionization from deployment in MEFO. The $NH_4^+$ mass spectrum is displayed as a left half-circle and the $H_3O^+$ is displayed as a right half-circle. Points are sized by the average signal across the campaign divided by the detection limit signal (DL: $3\sigma$ of campaign zero air background). The reagent-ion masses have been removed from the ion molecular mass. The top 100 ions in terms of signal-to-DL ratio for both reagent-ions have been colored according to their molecular formulae with periods of reagent-ion hysteresis removed (Sect. 3.4). A total of 725 ions are shown, and selected ions and their signal-to-DL ratios are displayed in a bar chart format in Fig. S9.

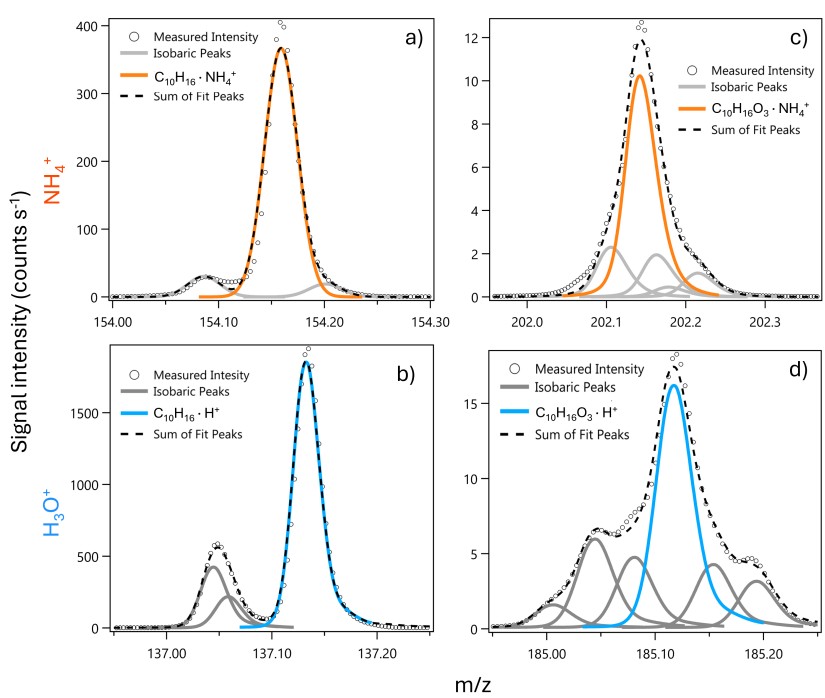

**Figure 8.** Selected campaign average high resolution mass spectra from MEFO for (a & b) monoterpene ($C_{10}H_{16}$) and (c & d) monoterpene oxygenate ($C_{10}H_{16}O_3$) molecular ions detected with $NH_4^+$ (a & c, orange) and $H_3O^+$ (b & d, blue).

### 3.5 Reagent-ion comparison from ambient measurements in MEFO

We compare the capabilities of $NH_4^+$ and $H_3O^+$ reagent-ions in a single instrument using ambient observations from MEFO (Sect. 2.5). Switching between $H_3O^+$ and $NH_4^+$ on a 15-minute timescale over the 21-day deployment allows us to directly compare the two reagent-ions in a predominantly biogenic environment (Hunter et al., 2017; Vermeuel et al., 2023a; Riches et al., 2024; Link et al., 2024). A single instrument approach avoids inlet and instrument design influences on detection that would otherwise complicate a direct reagent-ion comparison (e.g., Riva et al., 2019). Previously Zaytsev et al. (2019a) used a switching $NH_4^+/H_3O^+$ Ionicon PTR3 with a modified helical tripole reaction chamber to measure products from the ·OH initiated oxidation of 3-methylcatecol. This chamber study demonstrated the sets of compounds detected by each reagent-ion and concluded that $NH_4^+$ is able to detect larger, more functionalized molecules, while $H_3O^+$ is able to detect smaller organic molecules (Zaytsev et al., 2019a). To facilitate a direct and quantitative reagent-ion comparison, we use the ratio of campaign average ambient signal to detection limit signal (i.e., $3\sigma$ of the background) for each ion as a measure of the signal-to-noise ratio (Fig. 7). With the assumption that ambient concentrations measured with each reagent-ion over the campaign mean are equivalent, a higher signal-to-DL ratio also implies a lower detection limit. This analysis allows us to evaluate the relative capability of each reagent-ion without direct calibrations for multifunctional biogenic organic compounds (e.g., Hunter et al., 2017; Vermeuel et al., 2023a; Link et al., 2024).

$NH_4^+$ ionization detects oxygen-containing species with a higher signal-to-noise ratio than $H_3O^+$. At MEFO, four series of $C_xH_yO_z$ ions dominate our mass spectrum (highlighted in Fig. 7): $C_{10}H_{16}O_n$, $C_{10}H_{14}O_n$, $C_{10}H_{18}O_n$, and $C_5H_8O_n$ which represent a mixture of biogenic terpenoid compounds and their early generation oxidation products. The $C_{10}H_{16}O_n$ and $C_{10}H_{14}O_n$ series suggest a mixture of primary emissions, such as citral ($C_{10}H_{16}O$; $C^* = 1 \times 10^6$ µg m$^{-3}$) and thymol or carvone ($C_{10}H_{14}O$; $C^* = 1-9 \times 10^5$ µg m$^{-3}$) (McKinney et al., 2011; Kaser et al., 2013; Vermeuel et al., 2023a), and oxidation products of other terpenoids. $H_3O^+$ detects the reduced $C_{10}H_{14}$ species with higher signal-to-noise compared to $NH_4^+$, with signal-to-DL ratios of 10.5 and 6.00, respectively. Similarly, $H_3O^+$ detects $C_{10}H_{16}$ with a signal-to-DL ratio of 23.5, compared to 14.3 for $NH_4^+$. Following this $C_{10}H_{16}O_n$ series, the n = 1 ion is detected with a signal-to-DL ratio of 6.31 with $H_3O^+$ and 10.8 with $NH_4^+$. The tendency toward increased signal-to-noise with oxygenation for $NH_4^+$ ionization continues in the $C_{10}H_{14,16}O_n$ series up to $C_{10}H_{14,16}O_3$ (with $C^*$ between $4 \times 10^2$ and $9 \times 10^3$ µg m$^{-3}$, Table S7). The $C_{10}H_{18}O_n$ series lacks a $C_{10}H_{18}$ ion and the distribution in the x-y scatter between the $C_{10}H_{16}$ and $C_{10}H_{18}O$ peaks is bi-modal (Fig. S10), which suggests multiple paths to form $C_{10}H_{18}O$ ions. These paths are likely (1) primary emissions of $C_{10}H_{18}O$ terpenoid compounds with similar emission profiles to monoterpenes and (2) water clusters formed with monoterpenes ($C_{10}H_{18} \cdot H_2O \cdot H^+$ and $C_{10}H_{18} \cdot H_2O \cdot NH_4^+$). The $C_5H_8O_n$ series represents a combination of fragments, primary emissions and oxidation products. The $C_5H_8$ ion is likely a mixture of isoprene and fragments from larger oxygenates and 2,3,2-MBO with both reagent-ions (e.g., Kilgour et al., 2024). Within the $C_5H_8O_n$ series, the $C_5H_8$ signal-to-DL ratio is larger than expected with $NH_4^+$ ionization; we suspect that this is due partly to fragmentation of other compounds into the $C_5H_8$ mass. The $NH_4^+$ sensitivity to isoprene is very low; however 2,3,2-MBO fragments substantially into $C_5H_8$ through dehydration of the tertiary alcohol group (Fig. 3). Fragmentation into $C_5H_8$ combined with a low background (Fig. A3 & S11) leads to a very large signal-to-DL ratio

(Fig. 7). The $C_5H_8O_3$ and $C_5H_8O_4$ peaks are likely oxidation products of isoprene and 2,3,2-MBO, while $C_5H_8O_2$ is likely an isoprene oxidation product (Saunders et al., 2003; Jenkin et al., 2015).

$NH_4^+$ can detect organic nitrates that easily fragment with $H_3O^+$ (Aoki et al., 2007; Duncianu et al., 2017) and so often go undetected in ambient measurements with $H_3O^+$ ionization (Fig. 7). Organic nitrates ionized with $H_3O^+$ fragment to form nitric acid ($HNO_3$) or nitronium ions ($NO_2^+$), where the loss of $HNO_3$ results in fragmentation into the masses for other oxygenates (Aoki et al., 2007). The two predominant series of organic nitrate ions ($C_{10}H_{15}NO_n$ and $C_{10}H_{17}NO_n$ with $C^*$ down to tens of µg m$^{-3}$; Table S7) are generally detected with a higher signal-to-DL ratio with $NH_4^+$ ionization compared to $H_3O^+$ ionization, if the ion is detected with $H_3O^+$ at all (Fig. 7). The exception is $C_{10}H_{15}NO_3$ which is detected at higher signal-to-DL (2.13) with $H_3O^+$ ionization compared to $NH_4^+$ (0.774); though this ion may arise from dehydration of hydroxy nitrates (i.e., $C_{10}H_{17}NO_4$-$H_2O$) and that $H_3O^+$ is fragmenting larger organic nitrates into the $C_{10}H_{15}NO_3$ ion. The nitrates we observe are potentially a mixture of carbonyl, hydroxy, and peroxy nitrates derived from the oxidation of monoterpenes ($C_{10}H_{16}$) and potentially other terpenoid ($C_{10}H_{16}O$/$C_{10}H_{14}O$) precursors (Table S8, Fry et al., 2013; Jenkin et al., 2015; Faxon et al., 2018; Bates et al., 2022). Additionally, $C_5H_{11}NO_5$ and $C_5H_9NO_5$ are detected using both reagent-ions but both with higher signal-to-DL ratio with $NH_4^+$. $C_5H_{11}NO_5$ is likely a nitrate from the oxidation 2,3,2-MBO while both 2,3,2-MBO and isoprene could form the $C_5H_9NO_5$ ion (Link et al., 2024).

$NH_4^+$ ionization's ability to detect oxygenated compounds with higher signal-to-noise ratio and lower detection limits than $H_3O^+$ arises from two main factors. First, $NH_4^+$ is a softer ionization method compared to $H_3O^+$, resulting in less molecular ion fragmentation (Sect. 3.2). This is evident in our ambient data for the known $H_3O^+$ monoterpene fragment, $C_6H_8 \cdot H^+$ compared to the analogous fragment with $NH_4^+$ ionization, $C_6H_8 \cdot NH_4^+$. The fragment is present in $NH_4^+$ ionization mode at a 1:10 fragment-to-molecular-ion ratio, compared to a 1:1 ratio with $H_3O^+$ under our fIMR conditions (Fig. S10). Second, $NH_4^+$ has higher selectivity toward oxygenates compared to $H_3O^+$ (Sect. 3.2). Our ambient reagent-ion switching observations further demonstrate this selectivity (Fig. 8 & A3). The monoterpenes ($C_{10}H_{16}$) are easily distinguished from isobaric ions with both $H_3O^+$ and $NH_4^+$, but oxygenates ($C_{10}H_{16}O_3$) have multiple isobaric interferences with $H_3O^+$ (Fig. 8). Both higher selectivity and reduced fragmentation contribute to fewer isobaric ions with $NH_4^+$. While our observations demonstrate the utility of $NH_4^+$ for detecting oxidized species that $H_3O^+$ ionization struggles to detect (e.g., Yuan et al., 2017; Riva et al., 2019; Pagonis et al., 2019; Coggon et al., 2024), the extent of fragmentation for specific compounds is difficult to diagnose in ambient, and complex laboratory, mass spectra. Overall, our ambient reagent-ion comparison demonstrates quantitatively that $NH_4^+$ is complementary to $H_3O^+$, and together these two reagent-ions allow improved detection and identification of a range of biogenic reactive organic carbon compounds and their early generation oxidation products.

## 4   Conclusions

To expand the range of ROC detectable with a single chemical ionization instrument, we present an approach to combine two positive reagent-ions, $NH_4^+$ and $H_3O^+$, in a Vocus-CI-ToFMS. To accommodate the need for a constant ion-molecule reactor temperature during switching, we apply an $E/N$ space scanning approach to select fIMR conditions compatible with

both reagent-ions. We characterize the ability of $NH_4^+$ and $H_3O^+$ to detect a range of reduced and oxygenated VOCs and S/IVOCs through analysis of laboratory standards and find that $H_3O^+$ detects reduced species well and fragments function-alized oxygenates away from the molecular ion, while $NH_4^+$ retains the molecular ion and allows for improved detection of oxygenates. We find that fragmentation generally correlated with $E/N$, while sensitivity is impacted by a combination of ion transmission, competing ionization pathways, and molecular ion fragmentation. To diagnose and quantify the timescales for reagent-ion switching hystersis we compare the use of three ion-types: $NH_4^+$ reagent ions; a persistent ambient $NH_4^+$-adduct ion; and $NH_4^+$-adduct or proton-transfer molecular ions from an internal standard infused in the sampling inlet. Reagent-ion signal variability at each switch is driven largely by changes in ion transmission so is less representative of ion chemistry, while monitoring a product ion is more directly related to ionization reactions taking place in the fIMR. An internal standard signal provides the ideal means to monitor reagent-ion hysteresis with a known and persistent product ion; however, persistent ambient ions and internal standard product ions can produce similar rates of data retention ($\sim$86-89 % data retention across a full 1800 s switching cycle with a 0.05 % $s^{-1}$ rate-of-change threshold). We deploy our $NH_4^+/H_3O^+$ reagent-ion switch-ing Vocus-CI-ToFMS during a 3-week period at a rural pine forest (Manitou experimental forest observatory) to facilitate a direct and quantitative reagent-ion comparison. Our ambient observations demonstrate that $NH_4^+$ detects oxygenated ROC with higher signal-to-noise and lower DL, including organic nitrates that $H_3O^+$ does not detect, while $H_3O^+$ detects reduced species that are undetectable with $NH_4^+$. $NH_4^+/H_3O^+$ reagent-ion switching takes advantage of the complimentary nature of the two reagent-ions to expand the range of ROC detectable with a single instrument.

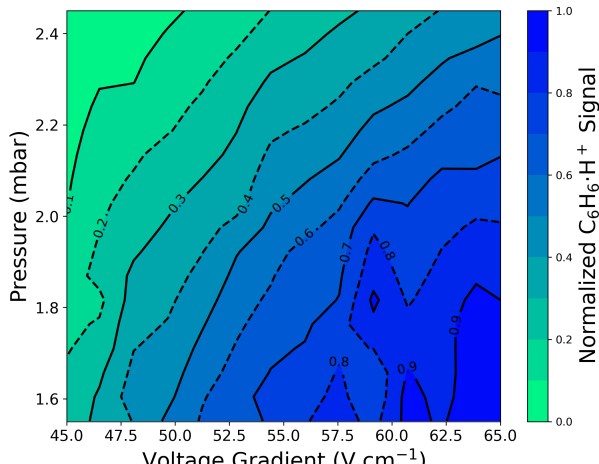

**Figure A1.** Normalized signal intensity for benzene measured with $H_3O^+$ ($C_6H_6 \cdot H^+$) from pressure-voltage gradient scans.

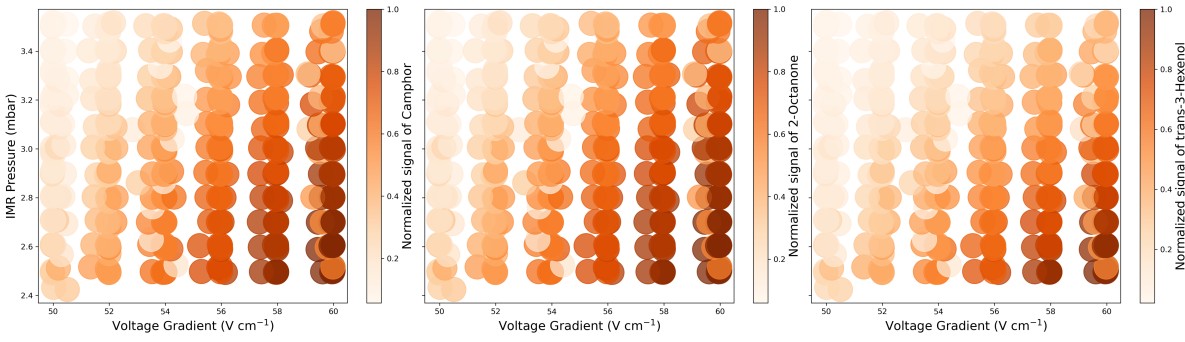

**Figure A2.** Normalized signal intensity for camphor (left), 2-octanone (middle), and trans-3-hexenol (right) measured with $NH_4^+$ ($C_{10}H_{16}O \cdot NH_4^+$, $C_8H_{16}O \cdot NH_4^+$, and $C_6H_{12}O \cdot NH_4^+$ respectively).

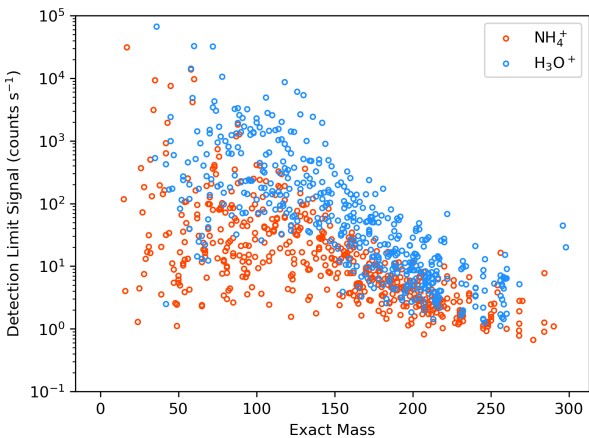

**Figure A3.** Mass dependent campaign average detection limit signal for ions detected with $NH_4^+$ (orange) and $H_3O^+$ (blue). The reagent-ion masses have been removed from ion exact masses.

*Author contributions.* CLZ & MDW designed research, CLZ collected and analyzed field and laboratory data with significant input from MDW, CLZ & MDW wrote the manuscript.

*Code and data availability.* Data and python code required to regenerate figures are available at https://doi.org/10.7910/DVN/FL0CZM (Zang and Willis, 2024).

*Competing interests.* The authors declare no competing interests.

*Acknowledgements.* This material is based on work supported in part by the US National Science Foundation under Grant No. AGS-2211153, and by Colorado State University (CSU). We thank the Flux Closure Study (FluCS) team with researchers from CSU, University of Minnesota-Twin Cities (UM), and Indiana University Bloomington (IU), including Sara Williams (CSU), Mj Riches (CSU), Matson Pothier (CSU), Michael Link (CSU), Lauren Garofalo (CSU), Delphine Farmer (CSU), Michael Vermeuel (UM), Dylan Millet (UM), Emily
Reidy (IU), Paige Price (IU), Brandon Bottorff (IU), and Phillip Stevens (IU), for their support with field observations at MEFO. We also acknowledge Tucker Melles (CSU) for supporting field observations at MEFO and ARTofMELT. We thank the National Center for Atmospheric Research (NCAR) for MEFO field site maintenance, and Paula Fornwalt and Steve Alton from United States Forest Service (USFS) for support during field observations. This work is part of the ARTofMELT (Atmospheric rivers and the onset of Arctic melt) project. The ARTofMELT expedition was supported and organized by the Swedish Polar Research Secretariat (SPRS) on the Swedish research icebreaker
Oden in spring 2023 under the SWEDARCTIC program. Support also came from the Swedish Council for Research Infrastructures (Grant 2021-00153) and the Knut and Alice Wallenberg Foundation (Grant 2016-0024). The authors are grateful to the co-Chief Scientists Michael Tjernström and Paul Zieger, the SPRS coordinator Åsa Lindgren and the SPRS support team, and to Captain Mattias Petersson and the crew on Oden.

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
