# Peer review of "Deployment and evaluation of an $NH_4^+/H_3O^+$ reagent-ion switching chemical ionization mass spectrometer for the detection of reduced and oxygenated gas-phase organic compounds"

_EGUsphere, 2024_

## Author Comment (AC1)

**Responses to Reviewer #1**

**General Comments:**

Zang & Willis present an instrument characterization study of a Vocus CIMS, switching between H3O+ and NH4+ reagent ions for the purpose of detecting and quantifying a large range of reactive organic carbon compounds with a single CIMS. They investigate and optimize their ion-molecule reactor conditions for each reagent, present methodologies for quantifying hysteresis timescales when switching reagents, and demonstrate their capabilities via ambient measurements of fresh and oxidized biogenic emissions. I believe this manuscript will serve as a solid foundation for new Vocus users who aim to use reagent switching. Many of my specific comments are meant to clarify details for such readers. I recommend this manuscript for publication following edits in response to the following comments.

**Specific Comments:**

Line 9 – Specify the integration time for the LODs (I believe 1 s?)

*The reviewer is correct that these LODs are given at 1Hz. This information has been added.*

Line 74 – The back reaction can also be important if the reaction is only slightly exothermic (e.g., HCHO in PTR).

*We appreciate the importance of this detail. We revised this sentence accordingly: "unless reaction timescales are long or the reaction is endothermic or only slightly exothermic."*

Line 110 – Specify the model of Vocus.

*This information (Vocus-S) has been added.*

Section 2.1 – Please include your fIMR and BSQ settings (amplitude and frequency) since they will also influence your sensitivities.

*We now include this information along with a full list of instrument settings as Table S1 in the supplement. Additionally we have added the fIMR and BSQ settings in sections 2.3, 2.5, and 2.6 in the revised manuscript.*

Line 127 – Rather than have the reader rely on the figures, list the full ranges of fIMR pressure and front voltage used in your experiments here as well. Also provide the corresponding, nominal E/N range.

*We have added two sentences at the end of Sect. 2.2 that read: "With $NH_4^+$, we characterized from 2.5 to 3.5 mbar and from 45 to 65 V cm-1 (60-120 Townsends (Td)). For $H_3O^+$, we characterized from 1.5 to 2.5 mbar and from 45 to 65 V cm-1 (80-200 Td)."*

Line 137 – Please provide the nominal E/N for each set of parameters.

*This information has been added, see our response to the previous comment.*

Section 2.3 – When calculating sensitivities, did you observe / account for interfering ions? For example, I find that the monoterpenes, including limonene, fragment to the C7H9+, toluene's quantitative ion.

*Our calibration cylinders are composed to avoid interference from fragmentation as much as possible, though we did not make this clear in the original manuscript. To address this we have included a table in the supplement (Table S2) with information on the three different cylinders used in Section 2.3. We have also added the following sentence: "The 23 analytes come from three separate multi-component cylinders where the composition was selected to avoid interferences from fragments (Table S2)." Specifically, monoterpenes and toluene are not in the same cylinder, so we avoid monoterpene fragmentation to $C_7H_9^+$ during calibration.*

Section 2.3 – A table in the SI with the observed fragments (and their abundances) would be useful for others attempting to characterize their own instrument.

*We agree entirely that this information is important to include. Table S6 has been added with information on fragmentation patterns shown in Fig. 3. We have also added the following to the Fig. 3 caption: "Data represented in this figure is shown in Table S6."*

Section 2.3 – Were any of the fragments you observed affected by the BSQ transmission attenuation? If so, did you correct for mass transmission when calculating fragmentation rates?

The smallest fragment we detect here is $C_5H_7^+$, which has an m/z of 67 and is above the BSQ mass filter range at 270V amplitude (Krechmer et al., 2018). For this reason we do not correct for mass transmission when calculating fragmentation rates.

Section 2.3 – In my experience, fragmentation seems to have a significant dependence on voltage gradients throughout the instrument (e.g., between the Vocus back voltage and the BSQ skimmer; or that same skimmer and the BSQ front voltage) in addition to the fIMR conditions. Have you investigated this dependence? If not, it may be prudent to note some of those gradients in your optimized setups for anyone attempting to recreate those conditions.

We agree entirely, and this information is now available in Table S1.

Section 2.3 – Was there a reason you settled on 60 °C (PTR is typically higher, 80-100 °C)? To promote NH4+ adducts? Higher temperatures are commonly used to limit adsorption, so would higher temperatures improve the hysteresis timescales?

Yes, we use 60 °C fIMR temperature to promote $NH_4^+$ adduct formation. We added a sentence acknowledging that this is different than traditional PTR conditions, and direct the reader to Section 3.1 where we discuss this choice in detail: *"Using a 60 °C reaction chamber with $H_3O^+$ is lower than commonly reported in the literature ($\sim$80-100 °C) (e.g., Vermeuel et al., 2023; Coggon et al., 2024); this choice arises from fIMR temperature constraints for $NH_4^+$ (Xu et al., 2022) and is discussed in more detail in Sect. 3.1."*

Line 170 – Which ion optics do you change? I don't believe there is any discussion of changing e.g., BSQ or PB settings in Section 3.1 or the methods. From line 315 ("... changes in the BSQ mass range") it sounds like there was a change, unless I'm misunderstanding.

We change a number of ion optics upon a switch in reagent-ion including the BSQ and other downstream voltages. This information is now included in Table S1.

Section 3.3 – You note humidity independence based on your results. Broadly, I agree. However, I highly encourage addition discussion of the minor humidity dependence at the highest humidities as shown in Fig. 4 (NH4+ $\sim$5% higher at 50% RH, $\sim$10% higher at 70+% RH – except for alkenes). NH4+ appears to have a stronger dependence than H3O+? Can you comment on compound-related trends?

We appreciate this suggestion and have now added the following sentences discussing the mild humidity dependencies with both reagent ions: *"Varying sample humidity with constant analyte concentration demonstrates low humidity dependence with both $NH_4^+$ and $H_3O^+$ ionization across a range of reduced and oxygenated ROC (Fig. 4). We note an approximately 10 % increase in the $NH_4^+$ sensitivity to nitriles and oxygenates while alkene sensitivities remain unchanged up to 85 % RH. We also observe a slight (5-10%) increase in sensitivity with humidity for oxygenated species with $H_3O^+$, while alkene sensitivities are less affected. The low humidity dependence of the Vocus-CI-ToFMS has been demonstrated previously for $H_3O^+$ for a variety of analytes (Krechmer et al., 2018; Kilgour et al., 2022; Li et al., 2024) and for a select number of small oxygenates, alkenes, and acetonitrile with $NH_4^+$ (Khare et al., 2022; Xu et al., 2022). We demonstrate the low dependence of sensitivity on sample humidity with $NH_4^+$ ionization under different instrumental conditions and for a selection of analytes including oxygenated alkenes and siloxanes (Fig. 4)."*

Line 318 – The analyte ion was chosen due to persistence, but does it provide a representative hysteresis timescale for most/all analytes? Have you attempted to repeat this process with other analytes and do they yield similar results? Are there other considerations readers should be aware of when picking analytes/internal standards?

Our intention in the using $C_3H_6O$-ammonium analyte ion was to demonstrate that this method is feasible given a persistent ambient ion signal is available. However, we did not intend to suggest that other users should necessarily apply this specific product ion to diagnose hysteresis. While we alluded to the pitfalls of this approach in the original manuscript, and noted that more ideal tracer ions would involve the use of an internal standard (Preprint Line 321: *"Despite this, future iterations of this approach would benefit from applying our method on the signal from an internal standard infused into the sampling inlet."*), we agree

entirely that this aspect of our manuscript should be strengthened. To do this, we have added analysis of ambient switching data from another campaign, in which we infused a deuterated internal standard mixture directly into our inlet. This revision necessitated including an additional figure (Fig. 6 in the revised manuscript, here as Fig. AC1) and the addition of a new methods Section 2.6 including information about this deployment. We have revised Section 3.4 into two sections, which focus on diagnosing the timescales of reagent-ion hysteresis when only reagent ions and potentially persistent ambient ions are available, and when an internal standard is available. Notably, the deployment for which we used an internal standard occurred in a marine environment, and as a result the ambient $C_3H_6O$-ammonium analyte ion is not persistent enough to allow a direct comparison with our analysis from MEFO. The hysteresis timescales of 2-hexanone-$d_4$ and the $C_3H_6O$-ammonium adduct are comparable between the two campaigns. For $NH_4^+$ ionization, 2-hexanone-$d_4$ has a hysteresis time of 34 s during ARTofMELT, compared to the $C_3H_6O$-ammonium timescale of 75 s during MEFO. For $H_3O^+$ ionization, 2-hexanone-$d_4$ has a hysteresis time of 168 s during ARTofMELT, compared the the $C_3H_6O$-ammonium timescale of 185 s during MEFO. Overall, major considerations for selecting a ion to diagnose reagent-ion hysteresis are: m/z above the BSQ mass filter, persistence and stability. In the revised Section 3.4 we have highlighted this set of key considerations.

[Figure]

Figure AC1: Ion signal after a reagent-ion switch for $NH_4^+$ (a & b) and $H_3O^+$ (c & d) in the ARTofMELT data, showing $NH_4 \cdot H_2O^+$ ions (a & c), $C_6H_8d_4O \cdot NH_4^+$ (b) and $C_6H_8d_4O \cdot H^+$ (d) internal standard ions. We grouped ion signals by the time after a switch and normalized the mean of each group by the maximum, and normalized means were fit with a bi-exponential function. The derivative of the fit ($\delta_n$) is displayed on the right axes (purple traces) and is used as a metric to filter reagent-ion hysteresis.

Line 319 – You mention that ambient variability may impact the derivation of hysteresis timescales. Have you performed the calculation in the absence of averaging (i.e., calculate the timescale for each reagent switch individually) to get a sense of the variability? If I were to apply the average hysteresis timescale to the whole campaign, is there a concern that some switches would have longer timescales that impact interpretability? We have incorporated an analysis of switch by switch variability in the supplement as Fig. S6 (Here as Fig. AC2). This analysis involved fitting traces after each switch with a bi-exponential fit and determining the time at which the derivative reached our 0.05 % s$^{-1}$ threshold. The results demonstrate that the mean cutoff from the analysis in Sect. 3.4 (dashed lines with dark background in Fig. AC2) does not entirely capture the variability in hysteresis timescales for the majority of individual switches. We have added the following discussion to address this: *"A switch-by-switch analysis of hysteresis from MEFO (available as Fig. S6) shows that the 0.05 % s$^{-1}$ $\delta_n$ cutoffs for $C_3H_6O \cdot NH_4^+$ calculated in Fig. 5b, d do not capture the majority of*

*the switch-by-switch cutoffs (37 % for $NH_4^+$ and 39 % for $H_3O^+$). Therefore, if a persistent ambient ion is used to diagnose hysteresis timescales, this should be done on a switch-by-switch basis. This variability may be associated with ambient variations in the $C_3H_6O·NH_4^+$ signal which can be avoided by applying our method described in Fig. 5 to a persistent and known signal from an internal standard (Sect. 3.4.2)."* Results of the switch-by-switch analysis differ when we use the signal of an internal standard to monitor hysteresis during ARTofMELT (Fig. AC1), where the cutoffs calculated in Fig AC1 for 2-hexanone-$d_4$ capture 75 % of the switch-by-switch variability for $NH_4^+$ and 72 % for $H_3O^+$. (Fig. AC3)

[Figure]

Figure AC2: Traces of bi-exponential fit for individual switches with $NH_4^+$ ionization (top) and $H_3O^+$ ionization (bottom) as well as a histogram of the time at which a 0.05 % $s^{-1}$ change threshold is reached across individual switches. Dashed lines with black background show the cutoff from the average analysis used in Sect. 3.4. Data is from the MEFO.

Line 323 – Do you have recommendations for reagent switching timescales? Can you comment on striking a balance between rapid switching to monitor more ROC vs longer dwell times to minimize the data loss to hysteresis?
We have used 15 minute switching during multiple deployments with this method, and this has allowed for reasonable data coverage with both reagent ions. However, this could be adjusted depending on the timescales relevant to different sampling applications. For an example the ARTofMELT analysis with a deuterated internal standard resulted in 4% data loss for $NH_4^+$ and 19% data loss for $H_3O^+$ during 15 minute switching. Since the time of hysteresis would likely remain relatively constant, if a user were to decrease the switching timescale to 5 minutes this would yield 12% data loss with $NH_4^+$ and 57% data loss with $H_3O^+$. Depending on the application, this trade-off off data coverage and time resolution might be appropriate. Alternatively, with 30 minute switches these reagent-ion hysteresis timescales would result in 2% data loss with $NH_4^+$ and 10% data loss for $H_3O^+$. Selection of switching timescales, and balancing data coverage and time-resolution, should be optimized based on the experiment. Our aim here is to provide a quantitative method to diagnosing these timescale such that future users can make informed decisions for their application.

Lines 324-325 & Fig. 5e – Figure 5e took a while to understand. I kept trying to compare it to the purple traces in Fig. 5a-d. I think additional explanation in Section 3.4 on how to interpret Figure 5e would be beneficial for the reader.
Our goal with Fig. 5e was to provide information on the impact of adjusting the $\delta_n$ threshold above and below 0.05 % s−1. Your comparison to the purple traces in Fig. 5a-d is justified, as Figure 5e is essentially

[Figure]

Figure AC3: Traces of bi-exponential fit for individual switches with $NH_4^+$ ionization (top) and $H_3O^+$ ionization (bottom) as well as a histogram of the time at which a 0.05 % $s^{-1}$ change threshold is reached across individual switches. Dashed lines with black background show the cutoff from the average analysis used in Sect. 3.4. Data is from a 2-week period of the ARTofMELT expedition.

depicting the purple traces from Fig. 5a-d on a 900 s timescale. Based on the reviewer's comment, we found this panel to be somewhat redundant and likely confusing for the reader. We have chosen to move Figure 5e to the supplement, and refer to it when we discuss the sensitivity of data loss to the magnitude of the normalized rate-of-change threshold.

Line 326 – Also provide the % data lost for each reagent, since it is asymmetric.
This information has been added.

Line 404-405 – Reiterate the threshold you used here. Also note the data retention for each ion chemistry due to the different timescales.
We have moved this discussion into Sect. 3.4 in the revised manuscript with additional details on the reagent-ion dependent data retention. We've included the following text in the conclusions section reiterating the threshold we used and the data retention: *"To diagnose and quantify the timescales for reagent-ion switching hystersis we compare the use of three ion-types: $NH_4^+$ reagent ions; a persistent ambient $NH_4^+$-adduct ion; and $NH_4^+$-adduct or proton-transfer molecular ions from an internal standard infused in the sampling inlet. Reagent-ion signal variability at each switch is driven largely by changes in ion transmission so is less representative of ion chemistry, while monitoring a product ion is more directly related to ionization reactions taking place in the fIMR. An internal standard signal provides the ideal means to monitor reagent-ion hysteresis with a known and persistent product ion; however, persistent ambient ions and internal standard product ions can produce similar rates of data retention (~86-89 % data retention across a full 1800 s switching cycle with a 0.05 % $s^{-1}$ rate-of-change threshold)."*

Figure 3 – The background shading is distracting. It took me a moment to realize they didn't represent data. I think the color coding of the standards' names is sufficient.
The background shading has now been removed from Fig. 3.

Figure 3 – I would typically associate "Molecular Ion Fraction" with A*H+ and A*NH4+ (i.e., inaccurate terminology for the fragments and clusters). Perhaps something closer to "Fractional Signal Contribution"?

We appreciate this suggestion and we have implemented the phrase "Fractional Signal Contribution" over Molecular ion fraction in the revised manuscript to be more general, as Figure 3 shows both molecular ions and fragment ions.

Figure 6 – How many ions are included in this plot?
There are a total of 725 ions in this figure. This information has been added to the figure legend of Fig. 7 in the revised manuscript.

Table S3 – Specify sensitivities are for NH4+.
The table caption now clarifies that these are sensitivities for $NH_4^+$.

Figure S4 – Expand the H3O+ panel y-axis so the dashed lines are more apparent.
The y-axis of this figure (which is Fig. S11 in the revised document manuscript) is now expanded.

**Technical Comments:**
Line 282 – Capitalize first word.
We have capitalized the first word to read "Reagent-ion".

**References**

M. M. Coggon, C. E. Stockwell, M. S. Claflin, E. Y. Pfannerstill, L. Xu, J. B. Gilman, J. Marcantonio, C. Cao, K. Bates, G. I. Gkatzelis, A. Lamplugh, E. F. Katz, C. Arata, E. C. Apel, R. S. Hornbrook, F. Piel, F. Majluf, D. R. Blake, A. Wisthaler, M. Canagaratna, B. M. Lerner, A. H. Goldstein, J. E. Mak, and C. Warneke. Identifying and correcting interferences to PTR-ToF-MS measurements of isoprene and other urban volatile organic compounds. *Atmospheric Measurement Techniques*, 17(2):801–825, Jan. 2024. ISSN 1867-8548. doi: 10.5194/amt-17-801-2024. URL `https://amt.copernicus.org/articles/17/801/2024/`.

P. Khare, J. E. Krechmer, J. E. Machesky, T. Hass-Mitchell, C. Cao, J. Wang, F. Majluf, F. Lopez-Hilfiker, S. Malek, W. Wang, K. Seltzer, H. O. T. Pye, R. Commane, B. C. McDonald, R. Toledo-Crow, J. E. Mak, and D. R. Gentner. Ammonium-adduct chemical ionization to investigate anthropogenic oxygenated gas-phase organic compounds in urban air. *Atmos Chem Phys*, 22(21):14377–14399, 2022. ISSN 1680-7316 (Print) 1680-7324 (Electronic) 1680-7316 (Linking). doi: 10.5194/acp-22-14377-2022. URL `https://www.ncbi.nlm.nih.gov/pubmed/36506646`.

D. B. Kilgour, G. A. Novak, J. S. Sauer, A. N. Moore, J. Dinasquet, S. Amiri, E. B. Franklin, K. Mayer, M. Winter, C. K. Morris, T. Price, F. Malfatti, D. R. Crocker, C. Lee, C. D. Cappa, A. H. Goldstein, K. A. Prather, and T. H. Bertram. Marine gas-phase sulfur emissions during an induced phytoplankton bloom. *Atmos Chem Phys*, 22:1601–1613, 2022. doi: https://doi.org/10.5194/acp-22-1601-2022.

J. Krechmer, F. Lopez-Hilfiker, A. Koss, M. Hutterli, C. Stoermer, B. Deming, J. Kimmel, C. Warneke, R. Holzinger, J. Jayne, D. Worsnop, K. Fuhrer, M. Gonin, and J. de Gouw. Evaluation of a new reagent-ion source and focusing ion–molecule reactor for use in proton-transfer-reaction mass spectrometry. *Analytical Chemistry*, 90(20):12011–12018, 2018. ISSN 0003-2700. doi: 10.1021/acs.analchem.8b02641. URL `https://doi.org/10.1021/acs.analchem.8b02641`.

F. Li, D. D. Huang, L. Tian, B. Yuan, W. Tan, L. Zhu, P. Ye, D. Worsnop, K. I. Hoi, K. M. Mok, and Y. J. Li. Response of protonated, adduct, and fragmented ions in vocus proton-transfer-reaction time-of-flight mass spectrometer (ptr-tof-ms). *Atmospheric Measurement Techniques*, 17(8):2415–2427, 2024. doi: 10.5194/amt-17-2415-2024. URL `https://amt.copernicus.org/articles/17/2415/2024/`.

M. P. Vermeuel, D. B. Millet, D. K. Farmer, M. A. Pothier, M. F. Link, M. Riches, S. Williams, and L. A. Garofalo. Closing the Reactive Carbon Flux Budget: Observations From Dual Mass Spectrometers Over a Coniferous Forest. *Journal of Geophysical Research: Atmospheres*, 128(14): e2023JD038753, July 2023. ISSN 2169-897X, 2169-8996. doi: 10.1029/2023JD038753. URL `https://agupubs.onlinelibrary.wiley.com/doi/10.1029/2023JD038753`.

L. Xu, M. M. Coggon, C. E. Stockwell, J. B. Gilman, M. A. Robinson, M. Breitenlechner, A. Lamplugh, J. D. Crounse, P. O. Wennberg, J. A. Neuman, G. A. Novak, P. R. Veres, S. S. Brown, and C. Warneke. Chemical ionization mass spectrometry utilizing ammonium ions (nh4+ cims) for measurements of organic compounds in the atmosphere. *Atmospheric Measurement Techniques*, 15(24):7353–7373, 2022. ISSN 1867-8548. doi: 10.5194/amt-15-7353-2022.

---

## Author Comment (AC2)

**Responses to Reviewer #2**

The authors presented a very nice characterization and application study of chemical ionization mass spectrometry (CIMS) using both ammonium and hydronium ions as reagent ions in the Vocus time-of-flight (ToF) CIMS. Automatic switching between these two reagents was realized and applied in a field campaign at a forested site. The results showed that although these two methods can detect most of VOCs studied (in total 23), hydronium ion (as in PTR-ToF-MS) is more suitable for reduced VOCs, while ammonium ion is more suitable for functionalized VOCs (S/IVOCs). A method was proposed to filter the periods with hysteresis during reagent ion switching, and field data were used to evaluate the performance of this method with auto-switching of reagent ions. The study is well designed and conducted, and the manuscript is clearly written. The findings are valuable in atmospheric chemistry research community in that it provides a method that can efficiently measure a wide range of VOCs and S/IVOCs in the atmosphere. I therefore recommend publication after Minor Revision, with some comments as follows.

**Main:**
The only concern I have is on the criteria of setting the time periods of data to filter out during the hysteresis due to reagent ion switching (section 3.4). In addition to the reagent ion (ammonium-water adduct), the author also chose the C3H6O-ammonium adduct to get the decreasing/increasing rates by taking the derivatives. And it seems that the authors preferred to use the results from the C3H6O-ammonium adduct to determine the time periods of data for filtering (comparing Figure 5 and L325). I have reservation on this for two reasons. First, it is indeed okay to assume that the constituents leading to the signal of C3H6O-ammonium adduct do not change within the 15 min of reagent ion switching. But one might not have good reasons to assume that the proportions of the C3H6O species in the air sampled (presumably acetone and propionaldehyde?) remain the same in the 550+ hours of data (2000+ decreasing/increasing curves used in Figure 5b and 5d). I assume that the ionization efficiency (or sensitivity) of acetone and propionaldehyde might differ with ammonium CIMS, or the authors can convince me otherwise. If so, and if their proportions in the C3H6O species changes, the shape of the exponential curves in Figure 5b and 5d) will be substantially distorted after averaging, thereby resulting in a high uncertainty in the estimation of the time for data filtering. The second reason is that by looking at Figure 5, the ammonium-water ion has both obvious exponential shape and its normalized signal intensity can restore to 1 in the switching from hydronium ion to ammonium ion (Figure 5c); the C3H6O-ammonium adduct ion, however, cannot restore even after 300 s (Figure 5d). Therefore, a better justification and clarification of choosing the data C3H6O-ammonium adduct ion instead of ammonium-water adduct ion to determine the time for data filtering are needed.

We agree with the reviewers concerns in utilizing $C_3H_6O$-ammonium adduct for monitoring switching due to changes in the isomer composition of ambient air sampled. The ammonium-water cluster does have a clear double exponential shape; however, the impacts of the BSQ on this signal make it so that the decay is dominated by the changing instrument voltages and it thus provides much less information about the ion chemistry taking place in the fIMR. Because the ammonium-water clusters are the reagent ions and we want to avoid hysteresis in ion chemistry, we argue that using an analyte ion is a more direct measure of ion chemistry relevant to analyte ion detection. However, as the reviewer points out, these reasons may not justify the assumptions required to use a persistent ambient ion. To address this shortcoming in our analysis, we provide additional data and associated analysis from another deployment (ARTofMELT) where we were able to deploy a deuterated internal standard constantly infused into our sampling inlet (Fig. 6 in the revised manuscript; see also our response to reviewer 1 comment related to Line 318 of the original manuscript). We monitor 2-hexanone-$d_4$ as a ammonium adduct and a proton-transfer product, and find that the average timescale for reagent-ion hysteresis is lower than that obtained with $C_3H_6O$-ammonium adduct at MEFO. While the $C_3H_6O$-ammonium adduct observed during ARTofMELT was too variable in time to allow a hysteresis analysis (shown in Fig. S7 of the revised manuscript), and so we cannot perform a perfectly direct comparison between the two deployments, our analysis does suggest that the hysteresis timescales obtained with $C_3H_6O$-ammonium adduct at MEFO are reasonable (i.e., a 75 s cutoff for $NH_4^+$, and a 175 s cutoff for $H_3O^+$).

For the field data, the authors only used the average mass spectra to compare the signal-to-noise ratios of

ammonium and hydronium ionization results. It would be good to show some comparison of selected species for both ionization methods to demonstrate the applicability of this method. That is, it is good to show that sensitivity of certain compound classes might different with different reagent ion, but the quantification results are still comparable. In addition, Figure 6 is a bit too difficult to distinguish the differences for those with small signal-to-noise ratios. It would be better to show results in bar charts for those series of compounds with different oxygen atoms that are generally in accordance with the discussion in the text (L346 onward).

We have added three figures to the supplement to address the reviewers suggestions. Fig. S9 shows bar charts for the species we discuss in Sect. 3.5 to help the reader visualize our discussion. Fig. S12 and Fig. S13 have been added to demonstrate what a time series and diurnal cycling looks like when using this reagent-ion switching method.

**Minor:**

L70: "A" should be no charge on it as a neutral analyte?

The reviewer is correct. This has been changed in the revised document.

L172-173: this looks like two sentences.

We agree the punctuation in this sentence was unclear. This now reads *"A change between ionization modes results in hysteresis where the ion chemistry is impure. The filtering of hysteretic periods is discussed in Sect. 3.4."*

L195: "evident from" or "different from"?

We used "evident from" here to suggest that the reduction in benzene sensitivity at lower $E/N$ was evidence of the production of protonated water clusters. We have adjusted the wording to make this distinction more clear for the reader. It now reads as follows: *"Second, at low $E/N$ protonated water clusters contribute to the ionization of $\alpha$-pinene. The production of protonated water clusters is evident from the reduced benzene sensitivity at lower $E/N$ (Fig. A1) (Gouw and Warneke, 2007)."*

L282: "Reagent-ion chemistry"?

This typo has been corrected.

Subsection title of 3.5: "Reagent-ion comparison"?

This typo has been corrected.

**References**

J. D. Gouw and C. Warneke. Measurements of volatile organic compounds in the earth's atmosphere using proton-transfer-reaction mass spectrometry. *Mass Spectrometry Reviews*, 26:223–257, 2007. ISSN 0277-7037. doi: 10.1002/mas.20119. URL `http://10.0.3.234/mas.20119` `https://dx.doi.org/10.1002/mas.20119`.

---

## Author Comment (AC3)

**Responses to Reviewer #3**

**General Comments:**
In this manuscript, Zang and Willis present the development and evaluation of a method for switching between the reagent ions, $NH_4^+$ and $H_3O^+$, in a Vocus-CI-ToFMS to detect reduced and oxygenated gas-phase reactive organic compounds (ROC). They detailed the optimization of ion-molecule reactor conditions for both reagent ions, compared their ability to detect a variety of ROC species, and applied the $NH_4^+/H_3O^+$ reagent-ion switching to the ambient measurements in a rural pine forest. Compared to CIMS measurements that employ complementary reagent ions either with repeated experiments or using different instruments, switching reagent ions in a single instrument as described in this study avoids the interferences from the changing inlets and instruments and enables a better evaluation of the capabilities of different regent ions in detecting ROC species. Overall, this work is solid and well designed, and the manuscript is nicely written. I recommend the publication of it in AMT after the following minor comments are addressed.

**Specific Comments:**
Line 127: Please also specify the default settings of fIMR pressure and front voltage here.
This information has been added as follows: *"With $NH_4^+$, we characterized from 2.5 to 3.5 mbar and from 45 to 65 V cm$^{-1}$ (60-120 Townsends (Td)). For $H_3O^+$, we characterized from 1.5 to 2.5 mbar and from 45 to 65 V cm$^{-1}$ (80-200 Td)."*

Line 140: Please provide the RH range evaluated here.
This information has been added.

Line 152: The ambient air was sampled using a 4-m long PFA tubing. Was the wall loss of ROC compounds significant in the sample inlet, especially for the oxygenated ROC?
We did not quantify the wall loss of ROC in the sampling inlet. We have added the following sentence to address this: *"The inlet likely produced wall loss of oxygenated ROC and while the extent was not quantified, minimizing the inlet inner diameter and maximizing the flow rate, while maintaining laminar flow, serve to minimize inlet losses and tubing delays (Pagonis et al., 2017)."*

Line 244: "Comparable" is not appropriate word here, as for many oxygenated ROC species shown in Figure 2, H3O+ exhibits significantly higher sensitivity than NH4+.
We have modified this statement to more accurately communicate what we intended with the word "comparable" here: *"For the compounds detected with both ionization modes, sensitivities and detection limits for $H_3O^+$ and $NH_4^+$ are in the same order of magnitude (Fig. 2 & A3)."*

Line 263: Not only 2-octanone, but also acetone. In Figure 3, only the molecular ion fraction of 2-hexanone is displayed for ketones. Suggest adding other ketones such as hydroxyacetone, methyl ethyl ketone, and methyl vinyl ketone to the figure, which also have a lower detection limit in $NH_4^+$ mode than in $H_3O^+$ mode.
We have noted the additional exception of acetone to this statement: *"This is consistent with the lower $NH_4^+$ detection limit for the majority of ketones we examined (with the exceptions of acetone and 2-octanone) and the aldehyde trans-2-hexenal (Fig. 2)."* Our intention with Fig. 3 is to display compounds that are complimentary to those displayed in Fig. 2 while providing insight into the general fragmentation patterns with $NH_4^+$ and $H_3O^+$ ionization for a range of compound classes and functional groups. Including the fragmentation patterns of smaller ketone species will therefore not add unique information to Fig. 3.

Line 264-265: The authors stated that "reduced fragmentation has a larger impact on sensitivity between the two reagent-ions for more highly oxidized compounds with multiple functional groups." However, the $H_3O^+$ ionization induces stronger fragmentation while having higher sensitivities to all ketone species, compared to the NH4+ ionization. Please modify this statement.
This statement was intended to apply to larger oxygenates that we discuss later in Sect. 3.5, but we acknowledge this was not clear. We have amended the sentence so that it clarifies that we are not referring to the ketone species in Fig. 2 and 3 as "highly oxidized compounds." The sentence now reads as: *"Our observations suggest that reduced fragmentation has a larger impact on detection capability of the two reagent-ions*

*for more highly oxidized compounds with multiple functional groups. This is observed for propane-1,2-diol which is readily detected with $NH_4^+$ but not with $H_3O^+$; the detection of oxidized ROC is discussed further in Sect. 3.5."*

Line 270: As shown in Figure 4, ion signals of several species show a small but noticeable positive dependence on the RH for both reagent ions. This phenomenon should be mentioned and the reason should be discussed. We appreciate this suggestion and have now added the following sentences discussing the mild humidity dependencies with both reagent ions: *"Varying sample humidity with constant analyte concentration demonstrates low humidity dependence with both $NH_4^+$ and $H_3O^+$ ionization across a range of reduced and oxygenated ROC (Fig. 4). We note an approximately 10 % increase in the $NH_4^+$ sensitivity to nitriles and oxygenates while alkene sensitivities remain unchanged up to 85 % RH. We also observe a slight (5-10%) increase in sensitivity with humidity for oxygenated species with $H_3O^+$, while alkene sensitivities are less affected. The low humidity dependence of the Vocus-CI-ToFMS has been demonstrated previously for $H_3O^+$ for a variety of analytes (Krechmer et al., 2018; Kilgour et al., 2022; Li et al., 2024) and for a select number of small oxygenates, alkenes, and acetonitrile with $NH_4^+$ (Khare et al., 2022; Xu et al., 2022)."*

Line 281 and Figure 5: Although the authors mentioned in the text that the influence of $H_3O^+$ reagent ions were not observed in $NH_4^+$ mode, it would be good to also plot the signal profiles of an example protonated ROC species in Figure a-e or in a separate figure.
A figure demonstrating this has been added to the supplement as Fig. S3.

Line 331: This comment is also related to Figure 5. As shown in Figure 5a, there remains a non-negligible fraction of $NH_4H_2O^+$ ion signals after switching the reagent ion to $H_3O^+$ for 300 s. Do the residual $NH_4H_2O^+$ ions in $H_3O^+$ mode contribute to the ionization and detection of the highly oxygenated ROC species that are undetectable with $H_3O^+$ in ambient measurements?
We do not observe the ionization of highly oxygenated ROC species by ammonium adduct ionization during $H_3O^+$ ionization periods. While it may appear to be a non-negligible fraction, there is a delay in data acquisition of approximately 10-30 seconds between each ionization mode, and as a result some instrument settings have already changed before data collection begins and some $NH_4^+ \cdot H_2O$ is already depleted. As a results, 100% signal intensity in $H_3O^+$ ionization mode is not equivalent to 100% signal intensity in $NH_4^+$ ionization mode in Figure 5.

**References**

P. Khare, J. E. Krechmer, J. E. Machesky, T. Hass-Mitchell, C. Cao, J. Wang, F. Majluf, F. Lopez-Hilfiker, S. Malek, W. Wang, K. Seltzer, H. O. T. Pye, R. Commane, B. C. McDonald, R. Toledo-Crow, J. E. Mak, and D. R. Gentner. Ammonium-adduct chemical ionization to investigate anthropogenic oxygenated gas-phase organic compounds in urban air. *Atmos Chem Phys*, 22(21):14377–14399, 2022. ISSN 1680-7316 (Print) 1680-7324 (Electronic) 1680-7316 (Linking). doi: 10.5194/acp-22-14377-2022. URL `https://www.ncbi.nlm.nih.gov/pubmed/36506646`.

D. B. Kilgour, G. A. Novak, J. S. Sauer, A. N. Moore, J. Dinasquet, S. Amiri, E. B. Franklin, K. Mayer, M. Winter, C. K. Morris, T. Price, F. Malfatti, D. R. Crocker, C. Lee, C. D. Cappa, A. H. Goldstein, K. A. Prather, and T. H. Bertram. Marine gas-phase sulfur emissions during an induced phytoplankton bloom. *Atmos Chem Phys*, 22:1601–1613, 2022. doi: https://doi.org/10.5194/acp-22-1601-2022.

J. Krechmer, F. Lopez-Hilfiker, A. Koss, M. Hutterli, C. Stoermer, B. Deming, J. Kimmel, C. Warneke, R. Holzinger, J. Jayne, D. Worsnop, K. Fuhrer, M. Gonin, and J. de Gouw. Evaluation of a new reagent-ion source and focusing ion–molecule reactor for use in proton-transfer-reaction mass spectrometry. *Analytical Chemistry*, 90(20):12011–12018, 2018. ISSN 0003-2700. doi: 10.1021/acs.analchem.8b02641. URL `https://doi.org/10.1021/acs.analchem.8b02641`.

F. Li, D. D. Huang, L. Tian, B. Yuan, W. Tan, L. Zhu, P. Ye, D. Worsnop, K. I. Hoi, K. M. Mok, and Y. J. Li. Response of protonated, adduct, and fragmented ions in vocus proton-transfer-reaction time-of-flight mass spectrometer (ptr-tof-ms). *Atmospheric Measurement Techniques*, 17(8):2415–2427, 2024. doi: 10.5194/amt-17-2415-2024. URL `https://amt.copernicus.org/articles/17/2415/2024/`.

D. Pagonis, J. E. Krechmer, J. de Gouw, J. L. Jimenez, and P. J. Ziemann. Effects of gas–wall partitioning in teflon tubing and instrumentation on time-resolved measurements of gas-phase organic compounds. *Atmospheric Measurement Techniques*, 10(12):4687–4696, 2017. ISSN 1867-8548. doi: 10.5194/amt-10-4687-2017.

L. Xu, M. M. Coggon, C. E. Stockwell, J. B. Gilman, M. A. Robinson, M. Breitenlechner, A. Lamplugh, J. D. Crounse, P. O. Wennberg, J. A. Neuman, G. A. Novak, P. R. Veres, S. S. Brown, and C. Warneke. Chemical ionization mass spectrometry utilizing ammonium ions (nh4+ cims) for measurements of organic compounds in the atmosphere. *Atmospheric Measurement Techniques*, 15(24):7353–7373, 2022. ISSN 1867-8548. doi: 10.5194/amt-15-7353-2022.